# PROGRESSIVE COMPRESSION WITH UNIVERSALLY QUANTIZED DIFFUSION MODELS

**Yibo Yang**\*  **Justus C. Will**\*  **Stephan Mandt**
Department of Computer Science
University of California, Irvine
{yibo.yang, jcwill, mandt}@uci.edu

## ABSTRACT

Diffusion probabilistic models have achieved mainstream success in many generative modeling tasks, from image generation to inverse problem solving. A distinct feature of these models is that they correspond to deep hierarchical latent variable models optimizing a variational evidence lower bound (ELBO) on the data likelihood. Drawing on a basic connection between likelihood modeling and compression, we explore the potential of diffusion models for progressive coding, resulting in a sequence of bits that can be incrementally transmitted and decoded with progressively improving reconstruction quality. Unlike prior work based on Gaussian diffusion or conditional diffusion models, we propose a new form of diffusion model with uniform noise in the forward process, whose negative ELBO corresponds to the end-to-end compression cost using universal quantization. We obtain promising first results on image compression, achieving competitive rate-distortion and rate-realism results on a wide range of bit-rates with a single model, bringing neural codecs a step closer to practical deployment. Our code can be found at `https://github.com/mandt-lab/uqdm`.

## 1 INTRODUCTION

A diffusion probabilistic model can be equivalently viewed as a deep latent-variable model (Sohl-Dickstein et al., 2015; Ho et al., 2020; Kingma et al., 2021), a cascade of denoising autoencoders that perform score matching at different noise levels (Vincent, 2011; Song & Ermon, 2019), or a neural SDE (Song et al., 2021b). Here we take the latent-variable model view and explore the potential of diffusion models for communicating information. Given the strong performance of these models on likelihood estimation (Kingma et al., 2021; Nichol & Dhariwal, 2021), it is natural to ask whether they also excel in the closely related task of data compression (MacKay, 2003; Yang et al., 2023).

Ho et al. (2020); Theis et al. (2022) first suggested a progressive compression method based on an unconditional diffusion model and demonstrated its strong potential for data compression. Such a *progressive* codec is desirable as it allows us to decode data reconstructions from partial bit-streams, starting from lossy reconstructions at low bit-rates to perfect (lossless) reconstructions at high bit-rates, all with a single model. The ability to decode intermediate reconstructions without having to wait for all bits to be received is a highly useful feature present in many traditional codecs, such as JPEG. The use of diffusion models has the additional advantage that we can, in theory, obtain perfectly realistic reconstructions (Theis et al., 2022), even at ultra-low bit-rates. Unfortunately, the proposed method requires the communication of Gaussian samples across many steps, which remains intractable because the exponential runtime complexity of channel simulation (Goc & Flamich, 2024).

In this work, we take first steps towards a diffusion-based progressive codec that is computationally tractable. The key idea is to replace Gaussian distributions in the forward process with suitable *uniform* distributions and adjust the reverse process distributions accordingly. These modifications allow the application of universal quantization (Zamir & Feder, 1992) for simulating uniform noise channels, avoiding the intractability of Gaussian channel simulation in (Theis et al., 2022).

Specifically, our contributions are as follows:

---

\*Equal contribution

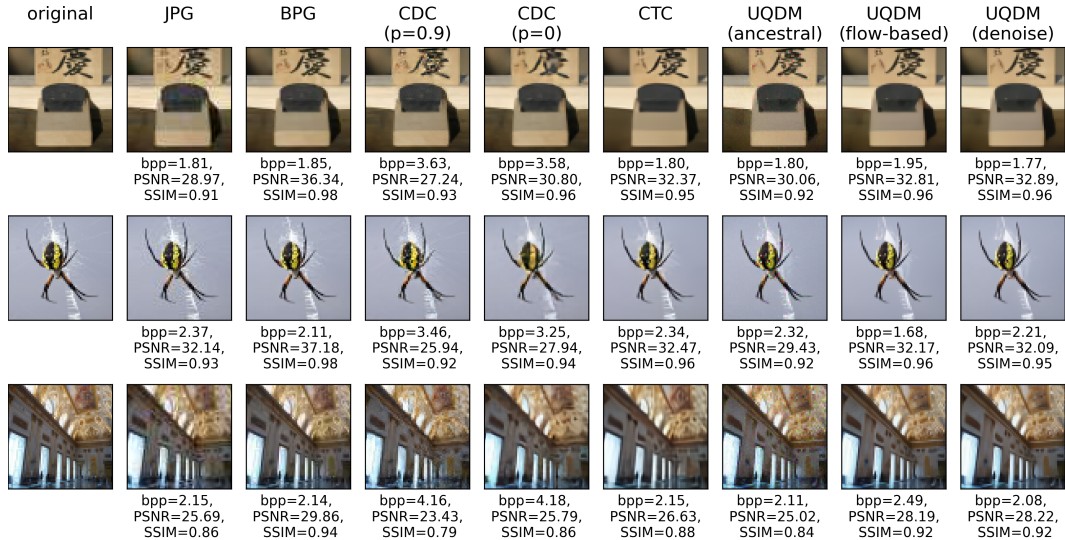

Figure 1: Example reconstructions from several traditional and neural codecs, chosen at roughly similar bitrates. At high bitrates, our UQDM method preserves details (e.g. shape and color pattern of the spider, or sharpness of the calligraphy) better than other neural codecs. Note that among the methods considered here, only ours and CTC (Jeon et al., 2023) implement progressive coding.

1. We introduce a new form of diffusion model, Universally Quantized Diffusion Model (UQDM), that is suitable for end-to-end learned progressive data compression. Unlike in the closely-related Gaussian diffusion model (Kingma et al., 2021), compression with UQDM is performed efficiently with universal quantization, avoiding the generally exponential runtime of relative entropy coding (Agustsson & Theis, 2020; Goc & Flamich, 2024).

2. We investigate design choices of UQDM, specifying its forward and reverse processes largely by matching the moments of those in Gaussian diffusion, and obtain the best results when we learn the reverse-process variance as inspired by Nichol & Dhariwal (2021).

3. We provide theoretical insight into UQDM in relation to VDM, and derive the continuous-time limit of its forward process approaching that of the Gaussian diffusion. These results may inspire future research in improving the modeling formalism and training efficiency.

4. We apply UQDM to image compression, and obtain competitive rate-distortion and rate-realism results which exceed existing progressive codecs at a wide range of bit-rates (up to lossless compression), all with a single model. Our results demonstrate, for the first time, the high potential of an unconditional diffusion model as a practical progressive codec.

## 2 BACKGROUND

**Diffusion models** Diffusion probabilistic models learn to model data by inverting a Gaussian noising process. Following the discrete-time setup of VDM (Kingma et al., 2021), the forward noising process begins with a data observation $\mathbf{x}$ and defines a sequence of increasingly noisy latent variables $\mathbf{z}_t$ with a conditional Gaussian distribution,

$$q(\mathbf{z}_t|\mathbf{x}) = \mathcal{N}(\alpha_t \mathbf{x}, \sigma_t^2 \mathbf{I}), \quad t = 0, 1, ..., T.$$

Here $\alpha_t$ and $\sigma_t^2$ are positive scalar-valued functions of time, with a strictly monotonically increasing *signal-to-noise-ratio* $\mathrm{SNR}(t) := \alpha_t^2/\sigma_t^2$. The *variance-preserving* process of DDPM (Ho et al., 2020) corresponds to the choice $\alpha_t^2 = 1 - \sigma_t^2$. The reverse-time generative model is defined by a collection of conditional distributions $p(\mathbf{z}_{t-1}|\mathbf{z}_t)$, a prior $p(\mathbf{z}_T) = \mathcal{N}(\mathbf{0}, \mathbf{I})$, and likelihood model $p(\mathbf{x}|\mathbf{z}_0)$. The conditional distributions $p(\mathbf{z}_{t-1}|\mathbf{z}_t) := q(\mathbf{z}_{t-1}|\mathbf{z}_t, \mathbf{x} = \hat{\mathbf{x}}_\theta(\mathbf{z}_t, t))$ are chosen to have the same distributional form as the "forward posterior" distribution $q(\mathbf{z}_{t-1}|\mathbf{z}_t, \mathbf{x})$, with $\mathbf{x}$ estimated from its noisy version $\mathbf{z}_t$ through the learned *denoising model* $\hat{\mathbf{x}}_\theta$. Further details on the forward and

backward processes can be found in Appendix A and B. Throughout the paper the logarithms use base 2. The model is trained by minimizing the negative ELBO (Evidence Lower BOund),

$$\mathcal{L}(\mathbf{x}) = \underbrace{\text{KL}(q(\mathbf{z}_T|\mathbf{x}) \, \| \, p(\mathbf{z}_T))}_{:=L_T} + \underbrace{\mathbb{E}\left[-\log p(\mathbf{x}|\mathbf{z}_0)\right]}_{:=L_{\mathbf{x}|\mathbf{z}_0}} + \sum_{t=1}^{T} \underbrace{\mathbb{E}\left[\text{KL}(q(\mathbf{z}_{t-1}|\mathbf{z}_t, \mathbf{x}) \, \| \, p(\mathbf{z}_{t-1}|\mathbf{z}_t))\right]}_{:=L_{t-1}}, \quad (1)$$

where the expectations are taken with respect to the forward process $q(\mathbf{z}_{0:T}|\mathbf{x})$. Kingma et al. (2021) showed that a larger $T$ corresponds to a tighter bound on the marginal likelihood $\log p(\mathbf{x})$, and as $T \to \infty$ the loss approaches the loss of a class of continuous-time diffusion models that includes the ones considered by Song et al. (2021b).

**Relative Entropy Coding (REC)**  Relative Entropy Coding (REC) deals with the problem of efficiently communicating a single sample from a target distribution $q$ using a coding distribution $p$. Suppose two parties in communication have access to a common "prior" distribution $p$ and pseudo-random number generators with a common seed; a Relative Entropy Coding (REC) method (Flamich et al., 2020) allows the sender to transmit a sample $\mathbf{z} \sim q$ using close to $\text{KL}(q \, \| \, p)$ bits on average. If $q$ arises from a conditional distribution, e.g., $q_\mathbf{x} = q(\mathbf{z} \mid \mathbf{x})$ is the inference distribution of a VAE (which can be viewed as a noisy *channel*), a *reverse channel coding* or *channel simulation* (Theis & Ahmed, 2022) algorithm then allows the sender to transmit $\mathbf{z} \sim q_\mathbf{x}$ with $\mathbf{x} \sim p(\mathbf{x})$ using close to $\mathbb{E}_{\mathbf{x} \sim p(\mathbf{x})}[\text{KL}(q(\mathbf{z} \mid \mathbf{x}) \, \| \, p(\mathbf{z}))]$ bits on average. At a high level, a typical REC method works as follows. The sender generates a (possibly large) number of candidate $\mathbf{z}$ samples from the prior $p$,

$$\mathbf{z}_n \sim p, \quad n = 1, 2, 3, ...,$$

and appropriately chooses an index $K$ such that $\mathbf{z}_K$ is a fair sample from the target distribution, i.e., $\mathbf{z}_K \sim q$. The chosen index $K \in \mathbb{N}$ is then converted to binary and transmitted to the receiver. The receiver recovers $\mathbf{z}_K$ by drawing the same sequence of $\mathbf{z}$ candidates from $p$ (made possible by using a pseudo-random number generator with the same seed as the sender) and stopping at the $K$th one.

A major challenge of REC algorithms is that their computational complexity generally scales exponentially with the amount of information being communicated (Agustsson & Theis, 2020; Goc & Flamich, 2024). As an example, the MRC algorithm (Cuff, 2008; Havasi et al., 2018) draws $M$ candidate samples and selects $K \in \{1, 2, , ..., M\}$ with a probability proportional to the importance weights, $q(\mathbf{z}_n)/p(\mathbf{z}_n), n = 1, ..., M$; similarly to importance sampling, $M$ needs to be on the order of $2^{\text{KL}(q\|p)}$ for $\mathbf{z}_K$ to be (approximately) a fair sample from $q$, thus requiring a number of drawn samples that scales exponentially with the relative entropy $\text{KL}(q\|p)$ (the cost of transmitting $K$ is thus $\log M \approx \text{KL}(q\|p)$ bits). The exponential complexity prevents, e.g., naively communicating the entire latent tensor $\mathbf{z}$ in a Gaussian VAE for lossy compression, as the relative entropy $\text{KL}(q(\mathbf{z}|\mathbf{x}) \, \| \, p(\mathbf{z}))$ easily exceeds thousands of bits, even for a small image. This difficulty can be partly remedied by performing REC on sub-problems with lower dimensions (Flamich et al., 2020; 2022) for which computationally viable REC algorithms exist (Flamich et al., 2024; Flamich, 2024), but at the expense of worse bitrate efficiency due to the accumulation of codelength overhead across the dimensions.

**Progressive Coding with Diffusion**  A *progressive* compression algorithm allows for lossy reconstructions with improving quality as more bits are sent, up till a lossless reconstruction. This results in variable-rate compression with a single bitstream, and is highly desirable in practical applications.

As we will explain, the NELBO of a diffusion model (eq. (1)) naturally corresponds to the *lossless* coding cost of a progressive codec, which can be optimized end-to-end on the data distribution of interest. Given a trained diffusion model, a REC algorithm, and a data point $\mathbf{x}$, we can perform progressive compression as follows (Ho et al., 2020; Theis et al., 2022): Initially, at time $T$, the sender transmits a sample of $q(\mathbf{z}_T|\mathbf{x})$ under the prior $p(\mathbf{z}_T)$, using $L_T$ bits on average. At each subsequent time step $t$, the sender transmits a sample of $q(\mathbf{z}_{t-1}|\mathbf{z}_t, \mathbf{x})$ given the previously transmitted $\mathbf{z}_t$, under the (conditional) prior $p(\mathbf{z}_{t-1}|\mathbf{z}_t)$, using approximately $L_{t-1}$ bits. Finally, given $\mathbf{z}_0$ at $t = 0$, $\mathbf{x}$ can be transmitted losslessly under the model $p(\mathbf{x}|\mathbf{z}_0)$ by an entropy coding algorithm (e.g., arithmetic coding), with a codelength close to $L_{\mathbf{x}|\mathbf{z}_0}$ bits (Polyanskiy & Wu, 2022, Chapter 13.1). Thus, the overall cost of losslessly compressing $\mathbf{x}$ sums up to $\mathcal{L}(\mathbf{x})$ bits, as in the NELBO in eq. (1). Crucially, at any time $t$, the receiver can use the most-recently-received $\mathbf{z}_t$ to obtain a *lossy* data reconstruction $\hat{\mathbf{x}}_t$. For this, several options are possible: Ho et al. (2020) consider using the diffusion model's denoising prediction $\hat{\mathbf{x}}_\theta(\mathbf{z}_t)$, while Theis et al. (2022) consider sampling $\hat{\mathbf{x}}_t \sim p(\mathbf{x}|\mathbf{z}_t)$, either by ancestral

sampling or a probability flow ODE (Song et al., 2021b). Note that if the reverse generative model captures the data distribution perfectly, then $\hat{\mathbf{x}}_t \sim p(\mathbf{x}|\mathbf{z}_t)$ follows the same marginal distribution as the data and has the desirable property of *perfect realism*, i.e., being indistinguishable from real data (Theis et al., 2022).

**Universal Quantization**   Although general-purpose REC algorithms suffer from exponential run-time (Agustsson & Theis, 2020; Goc & Flamich, 2024), efficient REC algorithms exist if we are willing to restrict the kinds of target and coding distributions allowed (Flamich et al., 2022; 2024). Here, we focus on the special case where the target distribution $q$ is given by a uniform noise channel, which is solved efficiently by Universal Quantization (UQ) (Roberts, 1962; Zamir & Feder, 1992; Agustsson & Theis, 2020). Specifically, suppose we (the sender) have access to a scalar r.v. $Y \sim p_Y$, and would like to communicate a noise-perturbed version of it,

$$\tilde{Y} = Y + U,$$

where $U \sim \mathcal{U}(-\Delta/2, \Delta/2)$ is an independent r.v. with a uniform distribution on the interval $[-\Delta/2, \Delta/2]$. UQ accomplishes this as follows: *Step 1.* Perturb $Y$ by adding another independent noise $U' \sim \mathcal{U}(-\Delta/2, \Delta/2)$, and quantize the result to the closet quantization point $K$ on a uniform grid of width $\Delta$, i.e., computing $K := \Delta \lfloor \frac{Y+U'}{\Delta} \rceil$ where $\lfloor \cdot \rceil$ denotes rounding to the nearest integer. *Step 2.* Entropy code and transmit $K$ under the conditional distribution of $K$ given $U'$. *Step 3.* The receiver draws the same $U'$ by using the same random number generator and obtains a reconstruction $\hat{Y} := K - U' = \Delta \lfloor \frac{Y+U'}{\Delta} \rceil - U'$. Zamir & Feder (1992) showed that $\hat{Y}$ indeed has the same distribution as $\tilde{Y}$, and the entropy coding cost of $K$ is related to the differential entropy of $\tilde{Y}$ via

$$H[K|U'] = I(Y; \tilde{Y}) = h(\tilde{Y}) - \log(\Delta).$$

In the above, the optimal entropy coding distribution $\mathbb{P}(K|U' = u')$ is obtained by discretizing $p_{\tilde{Y}} := p_Y \star \mathcal{U}(-\Delta/2, \Delta/2)$ on a grid of width $\Delta$ and offset by $U' = u'$ (Zamir & Feder, 1992), where $\star$ denotes convolution. If the true $p_{\tilde{Y}}$ is unknown, we can replace it with a surrogate density model $f_\theta(\tilde{y})$ during entropy coding and incur a higher coding cost,

$$\mathbb{E}_{y \sim P_Y}[\mathrm{KL}(u(\cdot|y) \,\|\, f_\theta(\cdot))] \geq I(Y; \tilde{Y}), \tag{2}$$

where $u(\cdot|y)$ denotes the density function of the uniform noise channel $q_{\tilde{Y}|Y=y} = \mathcal{U}(y - \Delta/2, y + \Delta/2)$. It can be shown that the optimal choice of $f_\theta$ is the convolution of $p_Y$ with $\mathcal{U}(-\Delta/2, \Delta/2)$. Therefore, as in prior work (Agustsson & Theis, 2020; Ballé et al., 2018), we will choose $f_\theta$ to have the form of another underlying density model $g_\theta$ convolved with uniform noise, i.e.

$$f_\theta(\cdot) = g_\theta(\cdot) \star \mathcal{U}(\cdot\,; -\Delta/2, \Delta/2). \tag{3}$$

## 3   UNIVERSALLY QUANTIZED DIFFUSION MODELS

We follow the same conceptual framework of progressive compression with diffusion models as in (Ho et al., 2020; Theis et al., 2022), reviewed in the previous section. While Theis et al. (2022) use Gaussian diffusion, relying on the communication of Gaussian samples which remains intractable in higher dimensions, we want to apply UQ to similarly achieve a compression cost given by the NELBO, while remaining computationally efficient. We therefore introduce a new model with a modified forward process and reverse process, which we term *universally quantized diffusion model* (UQDM), substituting Gaussian noise channels for uniform noise channels.

### 3.1   FORWARD PROCESS

The forward process of a standard diffusion model is often given by the transition kernel $q(\mathbf{z}_{t+1}|\mathbf{z}_t)$ (Ho et al., 2020) or perturbation kernel $q(\mathbf{z}_t|\mathbf{x})$ (Kingma et al., 2021), which in turn determines the conditional (reverse-time) distributions $q(\mathbf{z}_T|\mathbf{x})$ and $\{q(\mathbf{z}_{t-1}|\mathbf{z}_t, \mathbf{x})|t = 1, ..., T\}$ appearing in the NELBO in eq. (1). As we are interested in operationalizing and optimizing the coding cost associated with eq. (1), we will directly specify these conditional distributions to be compatible with UQ, rather than deriving them from a transition/perturbation kernel. We thus specify the forward process with

the same factorization as in DDIM (Song et al., 2021a) via $q(\mathbf{z}_{0:T}|\mathbf{x}) = q(\mathbf{z}_T|\mathbf{x}) \prod_{t=1}^{T} q(\mathbf{z}_{t-1}|\mathbf{z}_t, \mathbf{x})$, and consider a discrete-time non-Markovian process as follows,

$$
\begin{cases}
q(\mathbf{z}_T|\mathbf{x}) := \mathcal{N}(\alpha_T \mathbf{x}, \sigma_T^2 \mathbf{I}), \\
q(\mathbf{z}_{t-1}|\mathbf{z}_t, \mathbf{x}) := \mathcal{U}\left(b(t)\mathbf{z}_t + c(t)\mathbf{x} - \frac{\Delta(t)}{2}, b(t)\mathbf{z}_t + c(t)\mathbf{x} + \frac{\Delta(t)}{2}\right), t = 1, 2, ..., T,
\end{cases}
\tag{4}
$$

where $b(t)$, $c(t)$, and $\Delta(t)$ are scalar-valued functions of time. Note that unlike in Gaussian diffusion, our $q(\mathbf{z}_{t-1}|\mathbf{z}_t, \mathbf{x})$ is chosen to be a uniform distribution so that it can be efficiently simulated with UQ (as a result, our $q(\mathbf{z}_t|\mathbf{x})$ for any $t \neq T$ does not admit a simple distributional form). There is freedom in these choices of the forward process, but for simplicity we base them closely on the Gaussian case: we choose a standard isotropic Gaussian $q(\mathbf{z}_T|\mathbf{x})$, and set $b(t)$, $c(t)$, $\Delta(t)$ so that $q(\mathbf{z}_{t-1}|\mathbf{z}_t, \mathbf{x})$ has the same mean and variance as in the Gaussian case (see Appendix A for more details):

$$
b(t) = \frac{\alpha_t}{\alpha_{t-1}} \frac{\sigma_{t-1}^2}{\sigma_t^2}, \ c(t) = \sigma_{t|t-1}^2 \frac{\alpha_{t-1}}{\sigma_t^2}, \ \Delta(t) = \sqrt{12}\sigma_{t|t-1}\frac{\sigma_{t-1}}{\sigma_t}, \ \text{with } \sigma_{t|t-1}^2 := \sigma_t^2 - \frac{\alpha_t^2}{\alpha_{t-1}^2}\sigma_{t-1}^2.
$$

We note here that $q(\mathbf{z}_t|\mathbf{z}_T, \mathbf{x})$ can be written as a sum of uniform distributions, which as we increase $T \to \infty$, converges in distribution to a Gaussian by the Central Limit Theorem. This implies that $q(\mathbf{z}_t|\mathbf{x})$ also converges to a Gaussian for every $t$, and that our forward process has the same underlying continuous-time limit as in VDM (Kingma et al., 2021). We give the precise statement and a proof in Appendix A.3.

As in VDM (Kingma et al., 2021), the forward process schedules (i.e., $\alpha_t$ and $\sigma_t$, as well as $b(t), c(t), \Delta(t)$) can be learned end-to-end, e.g., by parameterizing $\sigma_t^2 = \text{sigmoid}(\phi(t))$, where $\phi$ is a monotonic neural network. We did not find this to yield significant improvements compared to using a linear noise schedule similar to the one in Kingma et al. (2021).

## 3.2 BACKWARD PROCESS

Analogously to the Gaussian case, we want to define a conditional distribution $p(\mathbf{z}_{t-1}|\mathbf{z}_t)$ that leverages a denoising model $\hat{\mathbf{x}}_t = \hat{\mathbf{x}}_\theta(\mathbf{z}_t, t)$ and closely matches the forward "posterior" $q(\mathbf{z}_{t-1}|\mathbf{z}_t, \mathbf{x})$. In our case, the forward "posterior" corresponds to a uniform noise channel with width $\Delta(t)$, i.e., $\mathbf{z}_{t-1} = b(t)\mathbf{z}_t + c(t)\mathbf{x} + \Delta(t)\mathbf{u}_t, \mathbf{u}_t \sim \mathcal{U}(-1/2, 1/2)$; to simulate it with UQ, we choose a density model for $\mathbf{z}_{t-1}$ with the same form as the convolution in eq. (3). Specifically, we let

$$
p(\mathbf{z}_{t-1}|\mathbf{z}_t) = g_\theta(\mathbf{z}_{t-1}; \mathbf{z}_t, t) \star \mathcal{U}(-\Delta(t)/2, \Delta(t)/2),
\tag{5}
$$

where $g_\theta(\mathbf{z}_{t-1}; \mathbf{z}_t, t)$ is a learned density chosen to match $q(\mathbf{z}_{t-1}|\mathbf{z}_t, \mathbf{x})$. Recall in Gaussian diffusion (Kingma et al., 2021), $p(\mathbf{z}_{t-1}|\mathbf{z}_t)$ is chosen to be a Gaussian of the form $q(\mathbf{z}_{t-1}|\mathbf{z}_t, \mathbf{x} = \hat{\mathbf{x}}_\theta(\mathbf{z}_t; t))$, i.e., the same as $q(\mathbf{z}_{t-1}|\mathbf{z}_t, \mathbf{x})$ but with the original data $\mathbf{x}$ replaced by a denoised prediction $\mathbf{x} = \hat{\mathbf{x}}_\theta(\mathbf{z}_t; t)$. For simplicity, we base $g_\theta$ closely on the choice of $p(\mathbf{z}_{t-1}|\mathbf{z}_t)$ in Gaussian diffusion, e.g.,

$$
g_\theta(\mathbf{z}_{t-1}; \mathbf{z}_t, t) = \mathcal{N}(b(t)\mathbf{z}_t + c(t)\hat{\mathbf{x}}_\theta(\mathbf{z}_t; t), \sigma_Q^2(t)\mathbf{I})
\tag{6}
$$

or a logistic distribution with the same mean and variance,

$$
g_\theta(\mathbf{z}_{t-1}; \mathbf{z}_t, t) = \text{Logistic}\left(b(t)\mathbf{z}_t + c(t)\hat{\mathbf{x}}_\theta(\mathbf{z}_t; t), \sigma_Q^2(t)\mathbf{I}\right).
\tag{7}
$$

where $\sigma_Q^2(t)$ is the variance of the Gaussian forward "posterior", and we use the same noise-prediction network for $\hat{\mathbf{x}}_\theta$ as in (Kingma et al., 2021). We found the Gaussian and logistic distributions to give similar results, but the logistic to be numerically more stable and therefore adopt it in all our experiments.

Inspired by (Nichol & Dhariwal, 2021), we found that learning a per-coordinate variance in the reverse process to significantly improve the log-likelihood, which we demonstrate in Sec. 5. In practice, this is implemented by doubling the output dimension of the score network to also compute a tensor of scaling factors $\mathbf{s}_\theta(\mathbf{z}_t)$, so that the variance of $g_\theta$ is $\boldsymbol{\sigma}_\theta^2 = \sigma_Q^2(t) \odot \mathbf{s}_\theta(\mathbf{z}_t)$. Refer to Appendix B.2 for a more detailed analysis of the log-likelihood and how a learned variance is beneficial.

We note that other possibilities for $g_\theta$ exist besides Gaussian or logistic, e.g., mixture distributions (Cheng et al., 2020), which trade off higher computation cost for increased modeling power. Analyzing the time reversal of the our forward process, similarly to (Song et al., 2021a), may also suggest better choices of the reverse-time density model $g_\theta$. We leave these explorations to future work.

We adopt the same form of categorical likelihood model $p(\mathbf{x}|\mathbf{z}_0)$ as in VDM (Kingma et al., 2021), as well as the use of Fourier features.

---

**Algorithm 1** Encoding

$\mathbf{z}_T \sim p(\mathbf{z}_T)$
**for** $t = T, \dots, 2, 1$ **do**
    Let $\Delta_t = \Delta(t)$, $\boldsymbol{\mu}_Q = b(t)\mathbf{z}_t + c(t)\mathbf{x}$.
    Compute the parameters of $p(\mathbf{z}_{t-1}|\mathbf{z}_t)$.
    ▷ *Send* $\mathbf{z}_{t-1} \sim q(\mathbf{z}_{t-1}|\mathbf{z}_t, \mathbf{x})$ *with UQ:*
    $\mathbf{u}_t \sim \mathcal{U}(-1/2, 1/2)$.
    $\mathbf{k}_t = \Delta_t \lfloor \frac{\boldsymbol{\mu}_Q}{\Delta_t} + \mathbf{u}_t \rceil$.
    Derive entropy model $p(\mathbf{k}|\mathbf{z}_t, \mathbf{u}_t)$ by discretizing $p(\mathbf{z}_{t-1}|\mathbf{z}_t)$.
    Entropy-encode $\mathbf{k}_t$ under $p(\mathbf{k}|\mathbf{z}_t, \mathbf{u}_t)$.
    $\mathbf{z}_{t-1} = \mathbf{k}_t - \Delta_t \mathbf{u}_t$.
**end for**
Entropy-encode $\mathbf{x}$ with $p(\mathbf{x}|\mathbf{z}_0)$.

---

**Algorithm 2** Decoding

$\mathbf{z}_T \sim p(\mathbf{z}_T)$     ▷ *Using shared seed*
**for** $t = T, \dots, 2, 1$ **do**
    Let $\Delta_t = \Delta(t)$.
    Compute the parameters of $p(\mathbf{z}_{t-1}|\mathbf{z}_t)$.
    $\mathbf{u}_t \sim \mathcal{U}(-1/2, 1/2)$.   ▷ *Using shared seed*
    Derive entropy model $p(\mathbf{k}|\mathbf{z}_t, \mathbf{u}_t)$ by discretizing $p(\mathbf{z}_{t-1}|\mathbf{z}_t)$.
    Entropy-decode $\mathbf{k}_t$ under $p(\mathbf{k}|\mathbf{z}_t, \mathbf{u}_t)$.
    $\mathbf{z}_{t-1} = \mathbf{k}_t - \Delta_t \mathbf{u}_t$.
    $\hat{\mathbf{x}}_t = \hat{\mathbf{x}}_\theta(\mathbf{z}_{t-1}; t-1)$.     ▷ *Lossy reconstruction*
**end for**
Entropy-decode $\mathbf{x}$ with $p(\mathbf{x}|\mathbf{z}_0)$.   ▷ *Lossless*

---

### 3.3 PROGRESSIVE CODING

Given a UQDM trained on the NELBO in eq. (1), we can use it for progressive compression similarly to (Ho et al., 2020; Theis et al., 2022), outlined in Section 2.

The initial step $t = T$ involves transmitting a Gaussian $\mathbf{z}_T$. Since we do not assume access to an efficient REC scheme for the Gaussian channel, we will instead draw the same $\mathbf{z}_T \sim p(\mathbf{z}_T) = \mathcal{N}(\mathbf{0}, \mathbf{I})$ on both the encoder and decoder side, with the help of a shared pseudo-random seed.[1] To avoid a train/compression mismatch, we therefore always ensure $q(\mathbf{z}_T|\mathbf{x}) \approx p(\mathbf{z}_T)$ and hence $L_T \approx 0$. At any subsequent step $t$, instead of sampling $\mathbf{z}_{t-1} = b(t)\mathbf{z}_t + c(t)\mathbf{x} + \Delta(t)\mathbf{u}'_t$ as in training, we apply UQ to communicate the "forward posterior" mean vector $\boldsymbol{\mu}_Q := b(t)\mathbf{z}_t + c(t)\mathbf{x}$. Specifically, given $\mathbf{z}_t$, the sender computes $\boldsymbol{\mu}_Q$ and the parameters of $p(\mathbf{z}_{t-1}|\mathbf{z}_t)$ (by evaluating the score network), draws a pseudo-random noise $\mathbf{u}_t \sim \mathcal{U}(-1/2, 1/2)$, quantizes $\boldsymbol{\mu}_Q$ to $\mathbf{k}_t = \Delta_t \lfloor \frac{\boldsymbol{\mu}_Q}{\Delta_t} + \mathbf{u}_t \rceil$ where $\Delta_t := \Delta(t)$, derives an entropy model $p(\mathbf{k}|\mathbf{z}_t, \mathbf{u}_t)$ (by discretizing $p(\mathbf{z}_{t-1}|\mathbf{z}_t)$ on a grid of width $\Delta_t$ and offset by $\mathbf{u}_t$), and entropy-encodes $\mathbf{k}_t$ under $p(\mathbf{k}|\mathbf{z}_t, \mathbf{u}_t)$. The receiver draws the same pseudo-random $\mathbf{u}_t \sim \mathcal{U}(-1/2, 1/2)$, entropy-decodes $\mathbf{k}_t$ under the same entropy model $p(\mathbf{k}|\mathbf{z}_t, \mathbf{u}_t)$, and computes $\mathbf{z}_{t-1} = \mathbf{k}_t - \Delta_t \mathbf{u}_t$ and (optionally) a lossy reconstruction $\hat{\mathbf{x}}_t$ from $\mathbf{z}_{t-1}$. Finally, after having transmitted $\mathbf{z}_0$ when $t = 1$, $\mathbf{x}$ is losslessly compressed using the entropy model $p(\mathbf{x}|\mathbf{z}_0)$. Pseudocode can be found in Algorithms 1 and 2. Note that we can replace the denoised prediction $\hat{\mathbf{x}} = \hat{\mathbf{x}}_\theta(\mathbf{z}_{t-1}; t-1)$ with more sophisticated ways to obtain lossy reconstructions such as flow-based reconstruction or ancestral sampling (Theis et al., 2022). As our method is progressive, the algorithm can be stopped at any time and the most recent lossy reconstruction be used as the output. Compared to compression with VDM (Theis et al., 2022), the main difference is that we transmit $\mathbf{z}_{t-1} \sim q(\mathbf{z}_{t-1}|\mathbf{z}_t, \mathbf{x})$ under $p(\mathbf{z}_{t-1}|\mathbf{z}_t)$ using UQ instead of Gaussian channel simulation; the overall computation complexity is now dominated by the evaluation of the denoising network $\hat{\mathbf{x}}_\theta$ (for computing the parameters of $p(\mathbf{z}_{t-1}|\mathbf{z}_t)$), which scales linearly with the number of time steps.

We implemented the progressive codec using `tensorflow-compression` (Ballé et al.), and found the actual file size to be within $3\%$ of the theoretical NELBO.

## 4 RELATED WORK

Diffusion models (Sohl-Dickstein et al., 2015) have achieved impressive results on image generation (Ho et al., 2020; Song et al., 2021a) and density estimation (Kingma et al., 2021; Nichol & Dhariwal, 2021). Our work is closely based on the latent-variable formalism of diffusion models (Ho et al.,

---

[1]This corresponds to a trivial REC problem where a sample from $q = p$ can be transmitted using $KL(q\|p) = 0$ bits.

2020; Kingma et al., 2021), with our forward and backward processes adapted from the Gaussian case. Our forward process is non-Markovian like DDIM (Song et al., 2021a), and our reverse process uses learned variance, inspired by (Nichol & Dhariwal, 2021). Recent research has focused on efficient sampling (Song et al., 2021a; Pandey et al., 2023) and better scalability via latent diffusion (Rombach et al., 2022), consistency models (Song et al., 2023), and distillation (Sauer et al., 2024), whereas we focus on the compression task. Related to our approach, cold diffusion (Bansal et al., 2024) showed that alternative forward processes other than the Gaussian still produce good image generation results.

Several diffusion-based neural compression methods exist, but they use conditional diffusion models (Yang & Mandt, 2023; Careil et al., 2023; Hoogeboom et al., 2023) which do not permit progressive decoding. Furthermore, they are also less flexible as a separate model has to be trained for each bitrate. Progressive neural compression has so far been mostly achieved by combining non-linear transform coding (for example using a VAE) with progressive quantization schemes. Such methods include PLONQ (Lu et al., 2021), which uses nested quantization, DPICT (Lee et al., 2022) and its extension CTC (Jeon et al., 2023), which use trit-plane coding, and DeepHQ (Lee et al., 2024) which uses a learned quantization scheme. Finally, codecs based on hierarchical VAEs (Townsend et al., 2024; Duan et al., 2023) are closely related but do not directly target the realism criterion.

# 5 EXPERIMENTS

We train UQDM end-to-end by directly optimizing the NELBO loss eq. (1), summing up $L_t$ across all time steps. This involves simulating the entire forward process $\{\mathbf{z}_0, ..., \mathbf{z}_T\}$ according to eq. (4) and can be computationally expensive when $T$ is large but can be avoided by using a Monte-Carlo estimate based on a single $L_t$ as in the diffusion literature (Ho et al., 2020). We found a small $T$ ($< 10$) to give the best compression performance, and therefore leave the investigation of training with a single-step Monte-Carlo objective to future work. Note that this would require sampling from the marginal distribution $q(\mathbf{z}_t|\mathbf{x})$, which becomes approximately Gaussian for large $t$ (see Sec. 3.1).

When considering the progressive compression performance of VDM and UQDM, we consider three ways of computing progressive reconstructions from $\mathbf{z}_t$: `denoise`, where $\hat{\mathbf{x}} = \hat{\mathbf{x}}_\theta(\mathbf{z}_t; t)$ is the prediction from the denoising network; `ancestral`, where $\hat{\mathbf{x}} \sim p(\mathbf{x}|\mathbf{z}_t)$ is drawn by ancestral sampling; and `flow-based` where $\hat{\mathbf{x}} \sim p(\mathbf{x}|\mathbf{z}_t)$ is computed deterministically using the probability flow ODE as in (Theis et al., 2022). In VDM, the probability flow ODE produces the same trajectory of marginal distributions as ancestral sampling, but gives improved lossy compression performance (Theis et al., 2022). In the case of UQDM, we apply the same update equations and observe similar benefits, likely due to the continuous-time equivalence of the underlying processes of UQDM and VDM. See Appendix B.3 for details. Note that DiffC-A and DiffC-F (Theis et al., 2022) directly correspond to our VDM results with `ancestral` and `flow-based` reconstructions.

In all experiments involving VDM and UQDM, we always use the same denoising U-net architecture for both, except UQDM uses twice as many output dimensions to additionally predict the reverse-process variance (see Sec. 3). We refer to Appendix Sec. C for further experiment details.

## 5.1 SWIRL TOY DATA

We obtain initial insights into the behavior of our proposed UQDM by experimenting on toy swirl data (see Appendix C.1 for details) and comparing with the hypothetical performance of VDM (Kingma et al., 2021).

First, we train UQDM end-to-end for various values of $T \in \{3, 4, 5, 10, 15, 20, 30\}$, with and without learning the reverse process variance. For comparison, we also train a single VDM with $T = 1000$, but compute the progressive-coding NELBO eq. (1) using different $T$. Fig. 2 plots the resulting NELBO values, corresponding to the bits-per-dimension cost of lossless compression. We observe that for UQDM, learning the reverse-process variance significantly improves the NELBO across all $T$, and a higher $T$ is not necessarily better. In fact, there seems to be an optimal $T \approx 5$, for which we obtain a bpd of around 8. The theoretical performance of VDM, by comparison, monotonically improves with $T$ (green curve) until it converges to a bpd of 5.8 at $T = 1000$, as consistent with theory (Kingma et al., 2021). We also tried initializing a UQDM without learned reverse-process variances to use the pre-trained VDM weights; interestingly, this resulted in very similar performance to the end-to-end trained result (blue curve), and further finetuning gave little to no improvement.

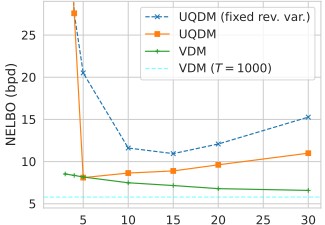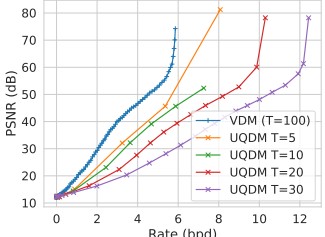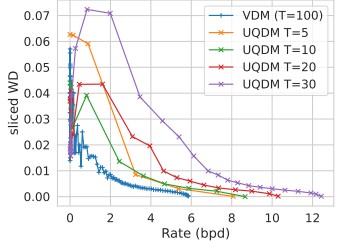

Figure 2: Results on swirl data. The VDM curves correspond to the hypothetical performance of REC that remains computationally intractable. **Left**: Lossless compression rates v.s. the choice of $T$, for UQDM with/without learned reverse-process variance (blue/orange) and VDM (green). For UQDM, learning the reverse-process variance significantly improved the NELBO, and an optimal $T \approx 5$. **Middle, Right**: Progressive lossy compression performance for VDM and UQDM, measured in fidelity (PSNR) v.s. bit-rate (middle), or realism (sliced Wasserstein distance) v.s. bit-rate (right).

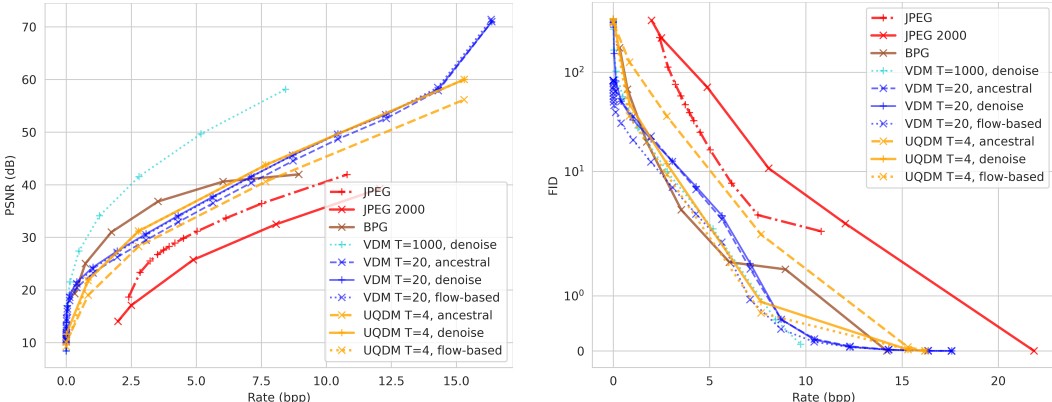

Figure 3: Progressive lossy compression performance of UQDM on the CIFAR10 dataset, comparing fidelity (PSNR) and realism (FID) with bit-rate per pixel (bpp), using either ancestral sampling or denoised prediction to obtain progressive reconstructions as indicated. The VDM curve corresponds to hypothetical performance of REC that is computationally intractable. We achieve better fidelity and realism than JPEG and JPEG2000 across all bit-rates and than BPG in the high bit-rate regime.

This suggests that a pretrained VDM can already be used for progressive compression with UQ via our moment-matching scheme (see Section 3), although the compression performance will be much worse compared to end-to-end trained UQDM with learned reverse-process variances.

We then examine the lossy compression performance of progressive coding. Here, we train UQDM end-to-end with learned reverse-process variances, and perform progressive reconstruction by ancestral sampling. Figure 2 plots the results in fidelity v.s. bit-rate and realism v.s. bit-rate. For reference, we also show the theoretical performance of VDM using $T = 100$ discretization steps, assuming a hypothetical REC algorithm that operates with no overhead. The results are consistent with those on lossless compression, with a similar performance ranking for $T$ among UQDM, and a gap remains to the hypothetical performance of VDM.

Finally, we examine the quality of unconditional samples from UQDM with varying $T$. Although our earlier results indicate worse compression performance for $T > 5$, Figure 7 shows that UQDM's sample quality monotonically improves with increasing $T$.

## 5.2 CIFAR10

Next, we apply our method to natural images. We start with the CIFAR10 dataset containing $32 \times 32$ images. We train a baseline VDM model with a smaller architecture than that used by Kingma et al. (2021), converging to around 3 bits per dimension. We use the noise schedule $\sigma_t^2 = \sigma(\gamma_t)$ where $\gamma_t$ is linear in $t$ with learned endpoints $\gamma_T$ and $\gamma_0$. For our UQDM model we empirically find that $T \approx 4$

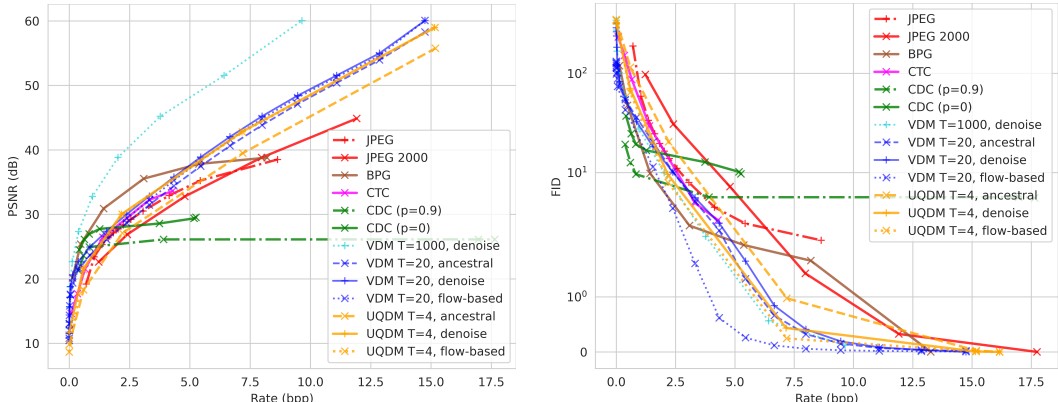

Figure 4: Progressive lossy compression performance of UQDM on the Imagenet64 dataset, comparing fidelity (PSNR) and realism (FID) with bit-rate per pixel (bpp), using either ancestral sampling or the denoised prediction to obtain progressive reconstructions as indicated. The VDM curve corresponds to hypothetical performance of REC that remains computationally intractable. While the reconstruction quality of other codecs like CDC or BPG plateaus at higher bit-rates, our method continues to gradually improve fidelity and realism even at higher bit-rates where it achieves the best results of any baseline. We beat compression performance of JPEG, JPEG2000, and CTC across all bit-rates. Note that only UQDM, CTC, and JPEG2000 implement progressive coding.

yields the best trade-off between bit-rate and reconstruction quality. We train our model end-to-end on the progressive coding NELBO eq. (1) with learned reverse-process variances.

We compare against the wavelet-based codecs JPEG, JPEG2000, and BPG (Bellard, 2018). For JPEG and BPG we use a fixed set of quality levels and encode the images independently, for JPEG2000 we instead use its progressive compression mode that allows us to set the approximate size reduction in each quality layer and obtain a rate-distortion curve from one bit-stream.

As shown in Figure 3, we consistently outperform both JPEG and JPEG2000 over all bit-rates and metrics. Even though BPG, a competitive non-progressive codec optimized for rate-distortion performance, achieves better reconstruction fidelity (as measured in PSNR) in the low bit-rate regime, our method closely matches BPG in realism (as measured in FID) and even beats BPG in PSNR at higher bit-rates. The theoretical performance of compression with Gaussian diffusion (e.g., VDM) (Theis et al., 2022), especially with a high number of steps such as $T = 1000$, is currently computationally infeasible, both due to the large number of neural function evaluations required, and due the intractable runtime of REC algorithms in the Gaussian case. Still, for reference we report theoretical results both for $T = 1000$ and $T = 20$, where the latter uses a smaller and more practical number of diffusion/progressive reconstruction steps.

## 5.3 IMAGENET $64 \times 64$

Finally, we present results on the ImageNet $64 \times 64$ dataset. We train a baseline VDM model with the same architecture as in (Kingma et al., 2021), reproducing their reported BPD of around $3.4$; we train a UQDM of the same architecture with learned reverse-process variances and $T = 4$. In addition to the baselines described in the previous section, we also compare with CTC (Jeon et al., 2023), a recent progressive neural codec, and CDC (Yang & Mandt, 2023), a non-progressive neural codec based on a conditional diffusion model that can trade-off between distortion and realism via a hyperparameter $p$. We separately report results for both $p = 0$, which purely optimizes the conditional diffusion objective, and $p = 0.9$, which prioritizes more realistic reconstructions that also jointly minimizes a perceptual loss. For CTC we use pre-trained model checkpoints from the official implementation (Jeon et al., 2023); for CDC we fix the architecture but train a new model for each bit-rate v.s. reconstruction quality/realism trade-off.

The results are shown in Figure 4. When obtaining progressive reconstructions from denoised predictions, UQDM again outperforms both JPEG and JPEG2000. Our results are comparable to, if not slightly better than, CTC, and even though the reconstruction quality of other codecs plateaus

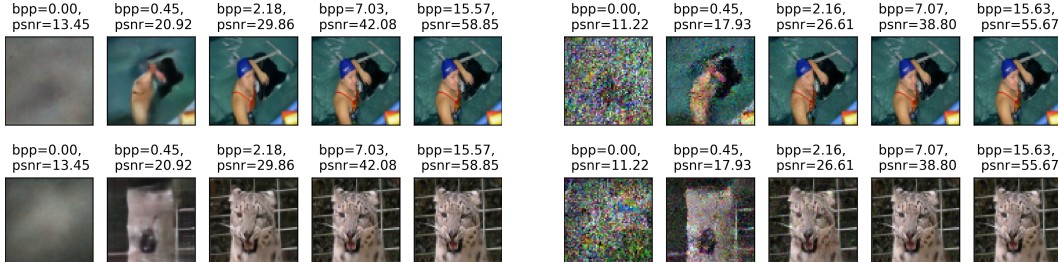

Figure 5: Example progressive reconstructions from UQDM trained with $T = 4$, obtained with denoised prediction (left) or ancestral sampling (right). The latter avoids blurriness but introduces graininess at low bit-rates, likely because the UQDM is unable to completely capture the data distribution and achieve perfect realism (perfect realism is also difficult to achieve also for Gaussian diffusion, as seen in the rate-realism plot of (Theis et al., 2022)). Flow-based reconstructions are qualitatively similar to the denoising-based reconstructions and can be found in Figure 8.

at higher bit-rates, our method continues to improve quality and realism gradually, even at higher bit-rates. Refer to Figures 1, 5 and 8 for qualitative results demonstrating progressive coding and comparison across codecs. At high bit-rates, UQDM preserves details better than other neural codecs. UQDM with denoised predictions tends to introduce blurriness, while ancestral sampling introduces graininess at low bit-rates, likely because the UQDM is unable to completely capture the data distribution and achieve perfect realism. Flow-based denoising matches the distortion of denoised predictions but achieves significantly higher realism as measured by FID. We note that the ideal of perfect realism (i.e., achieving 0 divergence between the data distribution and model's distribution) remains a challenge even for state-of-the-art diffusion models.

## 6 DISCUSSION

In this paper, we presented a new progressive coding scheme based on a novel adaptation of the standard diffusion model. Our universally quantized diffusion model (UQDM) implements the idea of progressive compression with an unconditional diffusion model (Theis et al., 2022) but bypasses the intractability of Gaussian channel simulation by using universal quantization (Zamir & Feder, 1992) instead. We present promising first results that match or outperform classic and neural compression baselines, including a recent progressive neural image compression method (Jeon et al., 2023). Given the practical advantages of a progressive neural codec – allowing for dynamic trade-offs between rate, distortion and computation, support for both lossy and lossless compression, and potential for high realism, all in a single model – our approach brings neural compression a step closer towards real-world deployment.

Future work may further improve our approach to close the performance gap to Gaussian diffusion; the latter represents the ideal lossy compression performance under a perfect realism constraint for an approximately Gaussian-distributed data source (Theis et al., 2022). This may require more sophisticated methods for computing progressive reconstructions that can achieve higher quality with fewer steps, or exploring different parameterizations of the forward and reverse processes with better theoretical properties. Finally, we expect further improvement in computation efficiency and scalability when combining our method with ideas such as latent diffusion (Rombach et al., 2022), distillation (Sauer et al., 2024), or consistency models (Song et al., 2023).

ETHICS STATEMENT

Our work focuses on the methodology of a learning-based data compression method, and thus has no direct ethical implications. The deployment of neural lossy compression however carries with it risks of miscommunication and misrepresentation (Yang et al., 2023), and needs to carefully analyzed and mitigated with future research.

REPRODUCIBILITY STATEMENT

We include proofs for all theoretical results introduced in the main text in Appendix A and B. We include further experimental and implementation details (including model architectures and other hyperparameter choices) in Appendix C. Our code can be found at https://github.com/mandt-lab/uqdm.

ACKNOWLEDGMENTS

Justus Will and Yibo Yang acknowledge support from the HPI Research Center in Machine Learning and Data Science at UC Irvine. Stephan Mandt acknowledges support from the National Science Foundation (NSF) under an NSF CAREER Award IIS-2047418 and IIS-2007719, the NSF LEAP Center, by the Department of Energy under grant DE-SC0022331, the IARPA WRIVA program, the Hasso Plattner Research Center at UCI, the Chan Zuckerberg Initiative, and gifts from Qualcomm and Disney. We thank Kushagra Pandey for feedback on the manuscript.

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

APPENDIX

## A   FORWARD PROCESS DETAILS

### A.1   GAUSSIAN (DDPM/VDM)

For completeness and reference, we restate the forward process and related conditionals given in (Kingma et al., 2021). The forward process is defined by

$$q(\mathbf{z}_t|\mathbf{x}) := \mathcal{N}(\alpha_t \mathbf{x}, \sigma_t^2 \mathbf{I}),$$

where $\alpha_t$ and $\sigma_t^2$ are positive scalar-valued functions of $t$. As in (Kingma et al., 2021), we define the following notation shorthand which are used in the rest of the appendix: for any $s < t$, let

$$\alpha_{t|s} := \frac{\alpha_t}{\alpha_s}, \quad \sigma_{t|s}^2 := \sigma_t^2 - \frac{\alpha_t^2}{\alpha_s^2}\sigma_s^2, \quad b_{t|s} := \frac{\alpha_t}{\alpha_s}\frac{\sigma_s^2}{\sigma_t^2}, \quad c_{t|s} := \sigma_{t|s}^2\frac{\alpha_s}{\sigma_t^2}, \quad \beta_{t|s} := \sigma_{t|s}\frac{\sigma_s}{\sigma_t}.$$

By properties of the Gaussian distribution, it can be shown that for any $0 \leq s < t \leq T$,

$$q(\mathbf{z}_t|\mathbf{z}_s) = \mathcal{N}(\alpha_{t|s}\mathbf{x}, \sigma_{t|s}^2\mathbf{I}),$$
$$q(\mathbf{z}_s|\mathbf{z}_t, \mathbf{x}) = \mathcal{N}(b_{t|s}\mathbf{z}_t + c_{t|s}\mathbf{x}, \beta_{t|s}^2\mathbf{I}),$$

In particular,

$$q(\mathbf{z}_{t-1}|\mathbf{z}_t, \mathbf{x}) = \mathcal{N}(b_{t|t-1}\mathbf{z}_t + c_{t|t-1}\mathbf{x}, \beta_{t|t-1}^2\mathbf{I}),$$
$$q(\mathbf{z}_t|\mathbf{z}_T, \mathbf{x}) = \mathcal{N}(b_{T|t}\mathbf{z}_t + c_{T|t}\mathbf{x}, \beta_{T|t}^2\mathbf{I}),$$

and we can use the reparameterization trick to write

$$\mathbf{z}_{t-1} = b_{t|t-1}\,\mathbf{z}_t + c_{t|t-1}\,\mathbf{x} + \beta_{t|t-1}\,\boldsymbol{\epsilon}_t, \ \boldsymbol{\epsilon}_t \sim \mathcal{N}(\mathbf{0}, \mathbf{I}),$$
$$\mathbf{z}_t = b_{T|t}\,\mathbf{z}_T + c_{T|t}\,\mathbf{x} + \beta_{T|t}\,\boldsymbol{\epsilon}_T, \ \boldsymbol{\epsilon}_T \sim \mathcal{N}(\mathbf{0}, \mathbf{I})$$

### A.2   UNIFORM (OURS)

Our forward process is specified by $q(\mathbf{z}_T|\mathbf{x})$ and $q(\mathbf{z}_{t-1}|\mathbf{z}_t, \mathbf{x})$ for each $t$, and closely follows that of the Gaussian diffusion. We set $q(\mathbf{z}_T|\mathbf{x})$ to be the same as in the Gaussian case, i.e.,

$$q(\mathbf{z}_T|\mathbf{x}) := \mathcal{N}(\alpha_T \mathbf{x}, \sigma_T^2 \mathbf{I}),$$

and $q(\mathbf{z}_{t-1}|\mathbf{z}_t, \mathbf{x})$ to be a uniform with the same mean and variance as in the Gaussian case, such that

$$q(\mathbf{z}_{t-1}|\mathbf{z}_t, \mathbf{x}) := \mathcal{U}(b_{t|t-1}\mathbf{z}_t + c_{t|t-1}\mathbf{x} - \sqrt{3}\beta_{t|t-1}, b_{t|t-1}\mathbf{z}_t + c_{t|t-1}\mathbf{x} + \sqrt{3}\beta_{t|t-1}),$$

or in other words,

$$\mathbf{z}_{t-1} = b_{t|t-1}\mathbf{z}_t + c_{t|t-1}\mathbf{x} + \sqrt{12}\beta_{t|t-1}\mathbf{u}_t, \quad \mathbf{u}_t \sim \mathcal{U}(-1/2, 1/2).$$

In the notation of eq. (4) this corresponds to letting $b(t) = b_{t|t-1}$, $c(t) = c_{t|t-1}$, $\Delta(t) = \sqrt{12}\beta_{t|t-1}$. It follows by algebraic manipulation that

$$\mathbf{z}_t = b_{T|t}\,\mathbf{z}_T + c_{T|t}\,\mathbf{x} + \underbrace{\sum_{v=t+1}^{T}\sqrt{12}\delta_{v|t}\mathbf{u}_v}_{:=\boldsymbol{\omega}_t}, \tag{8}$$

where

$$\mathbf{u}_v \sim \mathcal{U}(-1/2, 1/2), v = t+1, ..., T$$

are independent uniform noise variables, and

$$\delta_{v|t} := \beta_{v|v-1}\prod_{j=t+1}^{v-1}b_{j|j-1} = \frac{\sigma_t^2}{\alpha_t}\sqrt{\mathrm{SNR}(v-1) - \mathrm{SNR}(v)},$$

where

$$\mathrm{SNR}(s) := \frac{\alpha_s^2}{\sigma_s^2}.$$

It can be verified that

$$\mathbb{E}\left[\boldsymbol{\omega}_t\right] = \mathbf{0},$$

$$\mathrm{Var}\left(\boldsymbol{\omega}_t\right) = \sum_{v=t+1}^{T} \delta_{v|t}^2 \mathbf{I} = \frac{\sigma_t^4}{\alpha_t^2}[\mathrm{SNR}(t) - \mathrm{SNR}(T)]\mathbf{I} = \beta_{T|t}^2 \mathbf{I},$$

or in other words, at any step $t$ our forward-process "posterior" distribution $q(\mathbf{z}_t | \mathbf{z}_T, \mathbf{x})$ has the same mean and variance as in the Gaussian case.

## A.3 CONVERGENCE TO THE GAUSSIAN CASE

We show that both forward processes are equivalent in the continuous-time limit. To allow comparison across different number of steps $T$, we suppose that $\alpha_t$ and $\sigma_t$ are obtained from continuous-time schedules $\alpha(\cdot) : [0, 1] \to \mathbb{R}^+$ and $\sigma(\cdot) : [0, 1] \to \mathbb{R}^+$ (which were fixed ahead of time), such that $\alpha_t := \alpha(t/T)$ and $\sigma_t := \sigma(t/T)$ for $t = 0, \ldots, T$, for any choice of $T$. As in VDM (Kingma et al., 2021), we assume that the continuous-time signal-to-noise ratio $\mathrm{snr}(\cdot) := \alpha(\cdot)^2/\sigma(\cdot)^2$ is strictly monotonically decreasing.

To obtain the continuous-time limit, we hold the "continuous" time $\rho := \frac{t}{T}$ fixed for some $\rho \in [0, 1)$, and let $T \to \infty$ (or equivalently, let the time discretization $\frac{1}{T} \to 0$). We note that the quantities $b_{T|t}$, $c_{T|t}$, $\beta_{T|t}^2$ only depend on $\rho$, and are thus well-defined when we hold $\rho$ fixed and let $T \to \infty$:

$$b_{T|t} = \frac{\alpha_T}{\alpha_t} \frac{\sigma_t^2}{\sigma_T^2} = \frac{\alpha(1)}{\alpha(\rho)} \frac{\sigma^2(\rho)}{\sigma^2(1)},$$

$$c_{T|t} = \left(\sigma^2(1) - \frac{\alpha^2(1)}{\alpha^2(\rho)}\sigma^2(\rho)\right) \frac{\alpha(\rho)}{\sigma^2(1)},$$

$$\beta_{T|t}^2 = \left(\sigma^2(1) - \frac{\alpha^2(1)}{\alpha^2(\rho)}\sigma^2(\rho)\right) \frac{\sigma^2(\rho)}{\sigma^2(1)} = \frac{\sigma^4(\rho)}{\alpha^2(\rho)}(\mathrm{snr}(\rho) - \mathrm{snr}(1)).$$

We start by showing that our $q(\mathbf{z}_t | \mathbf{z}_T, \mathbf{x})$ converges to the corresponding Gaussian distribution in VDM in the continuous-time limit, which in turn implies the convergence of our $q(\mathbf{z}_t | \mathbf{x})$ to the corresponding Gaussian distribution in VDM.

**Theorem A.1.**
*For every fixed $\rho := \frac{t}{T} \in [0, 1)$, $q(\mathbf{z}_t | \mathbf{z}_T, \mathbf{x}) \xrightarrow{d} \mathcal{N}(b_{T|t}\,\mathbf{z}_T + c_{T|t}\,\mathbf{x}, \beta_{T|t}^2\,\mathbf{I})$ as $T \to \infty$.*

*Proof.*
Recall the following fact in the forward process of UQDM (see eq. (8)):

$$\mathbf{z}_t = b_{T|t}\,\mathbf{z}_T + c_{T|t}\,\mathbf{x} + \underbrace{\sum_{v=t+1}^{T} \sqrt{12}\delta_{v|t}\mathbf{u}_v}_{:=\boldsymbol{\omega}_t}, \tag{9}$$

where

$$\mathbf{u}_v \sim \mathcal{U}(-1/2, 1/2), v = t+1, \ldots, T$$

are independent uniform noise variables, and

$$\delta_{v|t} := \beta_{v|v-1} \prod_{j=t+1}^{v-1} b_{j|j-1} = \frac{\sigma_t^2}{\alpha_t}\sqrt{\mathrm{SNR}(v-1) - \mathrm{SNR}(v)},$$

where

$$\mathrm{SNR}(s) := \frac{\alpha_s^2}{\sigma_s^2}.$$

It therefore suffices to show that $\omega_t$ converges in distribution to $\mathcal{N}(\mathbf{0}, \beta_{T|t}^2 \mathbf{I})$ in the continuous-time limit. Since the different coordinates of $\omega_t$ are independent, we focus on a single coordinate and study the continuous-time limit of a scalar $\Omega_t$, given by a sum of scaled uniform variables,

$$\Omega_t := \sum_{v=t+1}^{T} \left( \frac{\sqrt{12}\sigma^2(\rho)}{\alpha(\rho)} \sqrt{\mathrm{snr}(\frac{v-1}{T}) - \mathrm{snr}(\frac{v}{T})} \right) U_v \tag{10}$$

$$= \sum_{j=1}^{n} \left( \frac{\sqrt{12}\sigma^2(\rho)}{\alpha(\rho)} \sqrt{\mathrm{snr}(\rho + \frac{j-1}{T}) - \mathrm{snr}(\rho + \frac{j}{T})} \right) U_j \tag{11}$$

where $U_j$'s are i.i.d. $\mathcal{U}(-1/2, 1/2)$ variables, and in the last step we set $n := n(T) = T - t$ and switched the summation index to $j = v - t$.

Define a triangular array of variables by

$$X_{n,j} = \left( \frac{\sqrt{12}\sigma^2(\rho)}{\alpha(\rho)} \sqrt{\mathrm{snr}(\rho + \frac{j-1}{T}) - \mathrm{snr}(\rho + \frac{j}{T})} \right) U_j,$$

for $j = 1, 2, ..., n$ and for $n \in \mathbb{N}^+$. For each $n$, $\{X_{n,j}\}_{j=1,2,...,n}$ are independent variables with $\mathbb{E}[X_{n,j}] = 0$, and it can be verified that

$$\sum_{j=1}^{n} \mathbb{E}[X_{n,j}^2] = \mathrm{Var}\,(\Omega_t) = \beta_{T|t}^2 = \frac{\sigma^4(\rho)}{\alpha^2(\rho)}(\mathrm{snr}(\rho) - \mathrm{snr}(1)).$$

To apply the Lindeberg-Feller central limit theorem (Durrett, 2019, Theorem 3.4.10) to $\Omega_t = X_{n,1} + ... + X_{n,n}$, it remains to verify the condition

$$\forall \epsilon > 0, \lim_{n \to \infty} \sum_{j=1}^{n} \mathbb{E}[X_{n,j}^2 \mathbf{1}\{|X_{n,j}| > \epsilon\}] = 0.$$

Let $\epsilon > 0$. Since $\mathrm{snr}(\cdot)$ is continuous on a compact domain $[0, 1]$, it is also uniformly continuous; then there exists a $\delta$ such that

$$|\mathrm{snr}(x_1) - \mathrm{snr}(x_2)| < \left( \frac{\epsilon\alpha(\rho)}{\sqrt{12}\sigma^2(\rho)} \right)^2, \quad \forall x_1, x_2, |x_1 - x_2| < \delta. \tag{12}$$

Let $T$ (and thus $n = T - t$) become sufficiently large such that $\frac{1}{T} < \delta$. Then, for all such $T$ (and thus $n$) sufficiently large, and for all $j$, it holds that $\mathbf{1}\{|X_{n,j}| > \epsilon\} = 0$ almost everywhere:

$$\mathbb{P}(|X_{n,j}| > \epsilon) = \mathbb{P}\left( \left( \frac{\sqrt{12}\sigma^2(\rho)}{\alpha(\rho)} \sqrt{\mathrm{snr}(\rho + \frac{j-1}{T}) - \mathrm{snr}(\rho + \frac{j}{T})} \right) |U_j| > \epsilon \right) \tag{13}$$

$$= \mathbb{P}\left( |U_j| > \frac{\epsilon\alpha(\rho)}{\sqrt{12}\sigma^2(\rho)} \frac{1}{\sqrt{\mathrm{snr}(\rho + \frac{j-1}{T}) - \mathrm{snr}(\rho + \frac{j}{T})}} \right) \tag{14}$$

$$\overset{\text{by eq. (12)}}{\leq} \mathbb{P}(|U_j| > 1) \tag{15}$$

$$= 0 \tag{16}$$

since $U_j \sim \mathcal{U}(-1/2, 1/2)$, and it follows that

$$\mathbb{E}[X_{n,j}^2 \mathbf{1}\{|X_{n,j}| > \epsilon\}] = 0$$

for all $j$ for all sufficiently large $n$. We conclude by the Lindeberg-Feller theorem that

$$\Omega_t = X_{n,1} + ... + X_{n,n} \xrightarrow{d} \mathcal{N}(0, \beta_{T|t}^2)$$

as $T \to \infty$. Applying the above argument coordinate-wise then proves the original statement. $\square$

**Corollary A.1.1.**

*If we assume $\sigma_T$ and $\alpha_T$ to be constants, then for every $t$, $q(\mathbf{z}_t|\mathbf{x}) \xrightarrow{d} \mathcal{N}(\alpha_t\mathbf{x}, \sigma_t^2\mathbf{I})$ as $T \to \infty$, that is, our forward model approaches the Gaussian forward process of VDM with an increasing number of diffusion steps.*

*Proof.* As $q(\mathbf{z}_T|x) = \mathcal{N}(\alpha_T\mathbf{x}, \sigma_T^2\mathbf{I})$ does not depend on $T$, the joint distribution $q(\mathbf{z}_t, \mathbf{z}_T|x) = q(\mathbf{z}_t|\mathbf{z}_T, \mathbf{x})q(\mathbf{z}_T|x)$ converges in distribution, which in turn implies convergence of $q(\mathbf{z}_t|x)$. The statement then follows from the identity

$$\mathcal{N}(\mathbf{z}_t; \alpha_t\mathbf{x}, \sigma_t^2\mathbf{I}) = \int \mathcal{N}(\mathbf{z}_t; b_{T|t}\,\mathbf{z}_T + c_{T|t}\,\mathbf{x}, \beta_{T|t}^2\,\mathbf{I})\,\mathcal{N}(\mathbf{z}_T; \alpha_T\mathbf{x}, \sigma_T^2\mathbf{I})\,d\mathbf{z}_T.$$

$\square$

# B  BACKWARD PROCESS DETAILS AND RATE ESTIMATES

## B.1  GAUSSIAN (DDPM/VDM)

Kingma et al. (2021) set $p(\mathbf{z}_{t-1}|\mathbf{z}_t) := q(\mathbf{z}_{t-1}|\mathbf{z}_t, \mathbf{x} = \hat{\mathbf{x}}_t) = \mathcal{N}(b_{t|t-1}\,\mathbf{z}_t + c_{t|t-1}\,\hat{\mathbf{x}}_t, \beta_{t|t-1}^2\mathbf{I})$ which yields

$$L_{t-1} = \mathrm{KL}(\mathcal{N}(b_{t|t-1}\,\mathbf{z}_t + c_{t|t-1}\,\mathbf{x}, \beta_{t|t-1}^2\mathbf{I}) \,\|\, \mathcal{N}(b_{t|t-1}\,\mathbf{z}_t + c_{t|t-1}\,\hat{\mathbf{x}}_t, \beta_{t|t-1}^2\mathbf{I}))$$

$$= \frac{1}{2}\frac{c_{t|t-1}^2}{\beta_{t|t-1}^2}\,\|\mathbf{x} - \hat{\mathbf{x}}_t\|_2^2 = \frac{1}{2}(\mathrm{SNR}(t-1) - \mathrm{SNR}(t))\,\|\mathbf{x} - \hat{\mathbf{x}}_t\|_2^2\,.$$

We have that $L_{t-1} \to 0$ as $T \to \infty$, due to the continuity of $\mathrm{SNR}(\cdot/T) = \mathrm{snr}(\cdot) = \alpha(\cdot)^2/\sigma(\cdot)^2$.

## B.2  UNIFORM (OURS)

Recall that we choose each coordinate of the reverse-process model $p(\mathbf{z}_{t-1}|\mathbf{z}_t)$ to have the density

$$p(\mathbf{z}_{t-1}|\mathbf{z}_t)_i := g_t(z) \star \mathcal{U}(z; -\Delta_t/2, \Delta_t/2)$$

$$= \frac{1}{\Delta_t}\int_{z-\Delta_t/2}^{z+\Delta_t/2} g_t(z)\,dz = \frac{1}{\Delta_t}(G_t(z + \Delta_t/2) - G_t(z - \Delta_t/2)),$$

where $G_t$ and $g_t$ are the cdf and pdf of a distribution with mean $\hat{\mu}_t := b_{t|t-1}z + c_{t|t-1}\hat{x}$ and variance $\sigma_g^2$, $z := (\mathbf{z}_t)_i$, $x := \mathbf{x}_i$, and $\hat{x} := \hat{\mathbf{x}}_\theta(\mathbf{z}_t; t)_i$. Using the shorthand $\mu_t := b_{t|t-1}z + c_{t|t-1}x$ we can derive the rate associated with the $i$th coordinate

$$L_{t-1} = \mathrm{KL}(\mathcal{U}(z; \mu_t - \Delta_t/2, \mu_t + \Delta_t/2) \,\|\, g_t(z) \star \mathcal{U}(z; -\Delta/2_t, \Delta/2_t))$$

$$= \frac{1}{\Delta_t}\int_{\mu_t - \Delta_t/2}^{\mu_t + \Delta_t/2} \log \frac{\frac{1}{\Delta_t}\mathbf{1}_{[\mu_t - \Delta_t/2, \mu_t + \Delta_t/2]}(z)}{\frac{1}{\Delta_t}(G_t(z + \Delta_t/2) - G_t(z - \Delta_t/2))}\,dz$$

$$= \frac{1}{\Delta_t}\int_{-\Delta_t/2}^{\Delta_t/2} \underbrace{-\log(G_t(z + \mu_t + \Delta_t/2) - G_t(z + \mu_t - \Delta_t/2))}_{:= h(z)}\,dz.$$

To gain some intuition for this rate, note that $h(z)$ is lowest when most of the probability mass of $G_t$ is concentrated tightly around $z + \mu_t$, which is the case when $|\mu_t - \hat{\mu}_t|$ is small. Specifically, if $G_t$ is in a distributional family with a standardized cdf $G_0$ such that $G_t(z) = G_0((z - \hat{\mu}_t)/\sigma_g)$ then

$$G_t(z + \mu_t + \Delta_t/2) - G_t(z + \mu_t - \Delta_t/2) \to \begin{cases} 1 & \text{if } |z + \mu_t - \hat{\mu}_t| < \Delta_t/2 \\ G_0(0) & \text{if } |z - \mu_t - \hat{\mu}_t| = \Delta_t/2 \\ 0 & \text{else} \end{cases}$$

as $\sigma_g \to 0$. Thus, if $|\mu_t - \hat{\mu}_t| \ll \Delta_t/2$, we obtain improved bit-rates for $\sigma_g$ that are small (relative to $\Delta_t$). On the other hand, as almost certainly $|\mu_t - \hat{\mu}_t| > 0$, we can't choose arbitrarily small

$\sigma_g$ because in that case both $\max(-h(-\Delta_t/2), -h(\Delta_t/2)) \to \infty$ and $L_{t-1} \to \infty$ as $\sigma_g \to 0$. This further motivates the merit of learning the backwards variances as $\sigma_g^2 = s_\theta(z)\beta_{t|t-1}^2 = s_\theta(z)\Delta_t^2/12$, allowing them to adapt to $|\mu_t - \hat{\mu}_t|$. Conversely, by the mean value theorem, there exists one $c \in (-\Delta_t/2, \Delta_t/2)$ so that

$$G_t(z + \mu_t + \Delta_t/2) - G_t(z + \mu_t - \Delta_t/2) = \Delta_t g_t(z + \mu_t + c) \approx \Delta_t g_t(z + \mu_t)$$

where the last approximation becomes more accurate for larger $\sigma_g$. If we further assume that $G_t$ is Gaussian (or sufficiently similar) $h(t)$ becomes approximately quadratic. In that case we study

$$h(z) \approx \left(1 - \frac{4z^2}{\Delta_t^2}\right)h(0) + \frac{2z^2 - \Delta_t z}{\Delta_t^2}h(-\Delta_t/2) + \frac{2z^2 + \Delta_t z}{\Delta_t^2}h(\Delta_t/2),$$

a quadratic function that exactly matches $h$ at values $z \in \{-\Delta_t/2, 0, \Delta_t/2\}$. Finally, this results in

$$L_{t-1} \approx \frac{1}{\Delta_t}\left[\frac{2}{\Delta_t^2}\left(h(-\Delta_t/2) + h(\Delta_t/2) - 2h(0)\right)\int_{-\Delta_t/2}^{\Delta_t/2} z^2\,dz + \frac{1}{\Delta_t}\left(h(\Delta_t/2) - h(\Delta_t/2)\right)\int_{-\Delta_t/2}^{\Delta_t/2} z\,dz + \Delta_t h(0)\right]$$
$$= -\frac{1}{6}\left[4h(0) + h(-\Delta_t/2) + h(\Delta_t/2)\right] \geq \frac{1}{3}\log(2),$$

where the last equality uses $h(z) \leq 0$ and $h(-\Delta_t/2) + h(\Delta_t/2) \leq \log(0.25)$ which follow from the fact that $G_t$ is a cdf. Empirically we note that this estimate is very accurate as long as $\sigma_g^2 \geq \beta_{t|t-1}^2$, demonstrating that simply matching moments as in VDM will occur a constant overhead for each diffusion step. As seen in Figure 2, this can be partly mitigated with smaller $\sigma_g^2$ but increasing the number of diffusion steps $T$ might still lead to an increase in ELBO. Numerical integration of $L_{t-1}$ confirms that if $\sigma_g^2$ is close to the optimal choice of $\sigma_g \approx |\mu_t - \hat{\mu}_t|$, $L_{t-1} \to 0$ as $T \to \infty$ as in the Gaussian case.

### B.3 Flow-based reconstructions

Given an intermediate latent $\mathbf{z}_t$, ancestral sampling yields an intermediate lossy reconstruction $\hat{\mathbf{x}} \sim p(\mathbf{x}|\mathbf{z}_t)$ that requires us to repeatedly sample from the conditional $p(\mathbf{z}_{t-1}|\mathbf{z}_t)$ until finally obtaining a reconstruction from $\mathbf{z}_0$ with the help of $p(\mathbf{x}|\mathbf{z}_0)$. This is equivalent to approximately solving a reverse SDE (Song et al., 2021c) and introduces additional noise during inference, which can make reconstructions grainy for diffusion models with a small number of steps, as can be seen in Figure 5. Song et al. (2021c) further note that an alternative approximate solution to the SDE can be obtained by deterministically reversing a "probability-flow" ODE (see also Theis et al. (2022)). Specifically, this involves repeatedly evaluating $\mathbf{z}_{t-1} = f(\mathbf{z}_t, t)$, where $f$ for VDM is defined as

$$f(\mathbf{z}_t, t) = \frac{\alpha_{t-1}}{\alpha_t}\mathbf{z}_t + \left(\sigma_{t-1} - \frac{\alpha_{t-1}}{\alpha_t}\sigma_t\right)\hat{\boldsymbol{\epsilon}}_t = \frac{\sigma_{t-1}}{\sigma_t}\mathbf{z}_t + \left(\alpha_{t-1} - \frac{\sigma_{t-1}}{\sigma_t}\alpha_t\right)\hat{\mathbf{x}}_t, \quad (17)$$

recovering the same process defined in (Song et al., 2021a). The equivalence of the continuous limit in Corollary A.1.1, suggests that the discrete-time backward processes of UQDM and VDM are similar enough in the sense that eq. (17) also approximately solves the implied reverse SDE of UQDM. Thus we use eq. (17) to obtain flow-based reconstructions for both VDM and UQDM.

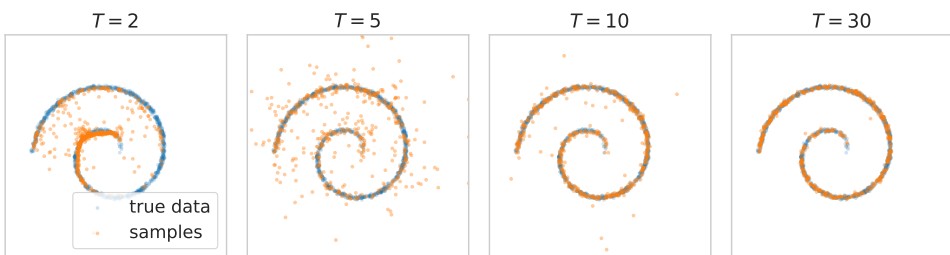

Figure 7: Unconditional samples from UQDM models trained with varying $T$ on the swirl dataset. The sample quality improves with larger $T$; however the compression performance becomes worse after $T > 5$, as discussed in Section 5.

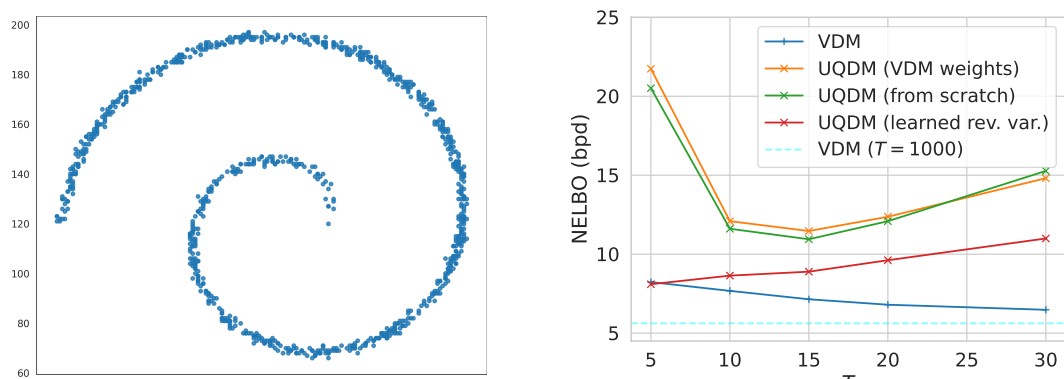

Figure 6: **Left:** 1000 samples from the toy swirl source. **Right:** Additional results on swirl data. We examined the compression performance of applying universal quantization to a pre-trained VDM model; conceptually this is equivalent to When using fixed reverse-process variances, we can directly re-use weights from a pretrained VDM model (orange), which achieves comparable results to training a UQDM model from scratch, even for a smaller number of timesteps.

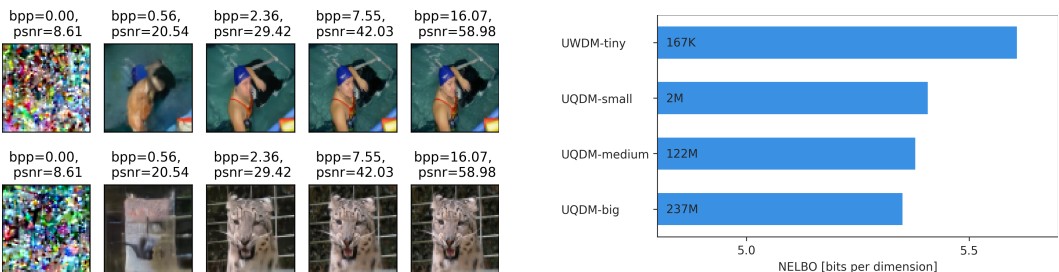

Figure 8: Additional results on ImageNet 64x64 data. **Left:** Example progressive reconstructions from UQDM trained with T = 4, obtained with flow-based denoising, as in Figure 5. Flow-based reconstructions achieve similar distortion (as meassured with PSNR) than denoised predictions at higher fidelity (as meassured with FID). **Right:** Ablation of the influence of model size on validation loss. Bars are labeled with the number of parameters for each model. Increasing the size of the denoising network allows for smaller bitrates.

## C    ADDITIONAL EXPERIMENTAL RESULTS

### C.1    SWIRL DATA

We use the swirl data from the codebase of (Kingma et al., 2021); Figure 6 shows 1000 samples from the toy data source. We use the same denoisng network $\hat{x}_\theta$ as in the official implementation,[2] which consists of 2 hidden layers with 512 units each. Figure 6 highlights the consequence of Corollary A.1.1: Because VDM and UQDM share the same continuous limit, we can use the weights of a pretrained VDM to obtain comparable UQDM results as a UQDM model that has been trained from scratch.

### C.2    CIFAR10

We use a scaled-down version of the denoising network from the VDM paper (Kingma et al., 2021) for faster experimentation. We use a U-Net of depth 8, consisting of 8 ResNet blocks in the forward direction and 9 ResNet blocks in the reverse direction, with a single attention layer and two additional ResNet blocks in the middle. We keep the number of channels constant throughout at 128.

We verified that our UQDM implementation based on `tensorflow-compression` achieves file size close the theoretical NELBO. When compressing a single 32x32 CIFAR image, we observe file size overhead $\leq 3\%$ of the theoretical NELBO. In terms of computation speed, it takes our model with fixed reverse-process variance less than 1 second to encode or decode a CIFAR image, either on CPU or GPU,[3] likely because the very few neural-network evaluations required ($T = 4$). For our model with learned reverse-process variance, however, it takes about 5 minutes to compress or decompress a CIFAR image, with nearly all of the compute time spent on a single CPU core. This is because with learned reverse-process variance, each latent dimension has a different predicted variance, and a separate CDF table needs to be built for each latent dimension during entropy coding; the `tensorflow-compression` library builds the CDF table for each coordinate in a naive for-loop rather than in parallel. Thus we expect the coding speed to be dramatically faster with a parallel implementation of entropy coding, e.g., using the `DietGPU`[4] library.

### C.3    IMAGENET $64 \times 64$

We use the same denoising network as in the VDM paper (Kingma et al., 2021). We use a U-Net of depth 64, consisting of 64 ResNet blocks in the forward direction and 65 ResNet blocks in the reverse direction, with a single attention layer and two additional ResNet blocks in the middle. We keep the number of channels constant throughout at 256. To investigate the impact of the size of the denoising network, in addition to this configuration with 237M parameters we call UQDM-big, we also run experiments with three smaller networks with 32 ResNet blocks and 128 channels (UQDM-medium, 122M parameters), 8 ResNet blocks and 64 channels (UQDM-small, 2M parameters), and 1 ResNet block and 32 channels (UQDM-tiny, 127K parameters), respectively. Smaller network are significantly faster and more resource-efficient but will naturally suffer from higher bitrates, as can be seen in Figure 8.

The required number of FLOPS per pixel for encoding and decoding is strongly dominated by the number of neural function evaluations (NFE) of our denoising network which depends on how soon we stop the encoding and decoding process. For lossless compression we have to multiple the FLOPS per NFE with $T$ which is equal to 4 in our case. For lossy compression after $t$ steps, with lossy reconstructions obtained through a denoised prediction, we obtain the required FLOPS for encoding and decoding by multiplying with $t$ and $t + 1$ respectively. The FLOPS per NFE depend on the network size, our investigated model size require 389K, 2.3M, 105M, and 204M FLOPS per pixel, in order from smallest to biggest model.

---

[2]`https://github.com/google-research/vdm/blob/main/colab/2D_VDM_Example.ipynb`
[3]Around 0.6 s for encoding and 0.5 s for decoding on `Intel(R) Xeon(R) Gold 5218 CPU @ 2.30GHz` CPU; 0.5 s for encoding and 0.3 s for decoding on a single `Quadro RTX 8000` GPU.
[4]`https://github.com/facebookresearch/dietgpu`

Figures 9 and 10 show more example reconstructions from several traditional and neural codecs, similar to Figure 1. At lower bitrates the artifacts each compression codecs introduces become more visible.

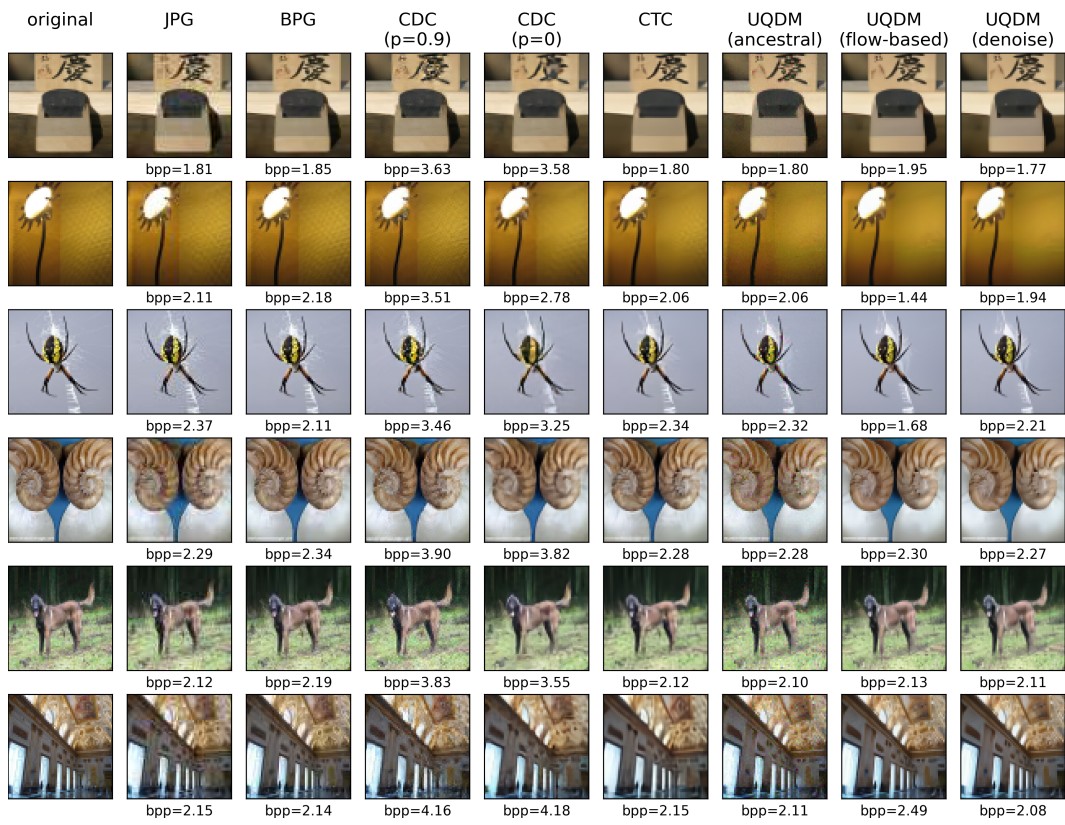

Figure 9: Additional example reconstructions , chosen at roughly similar (high) bitrates.

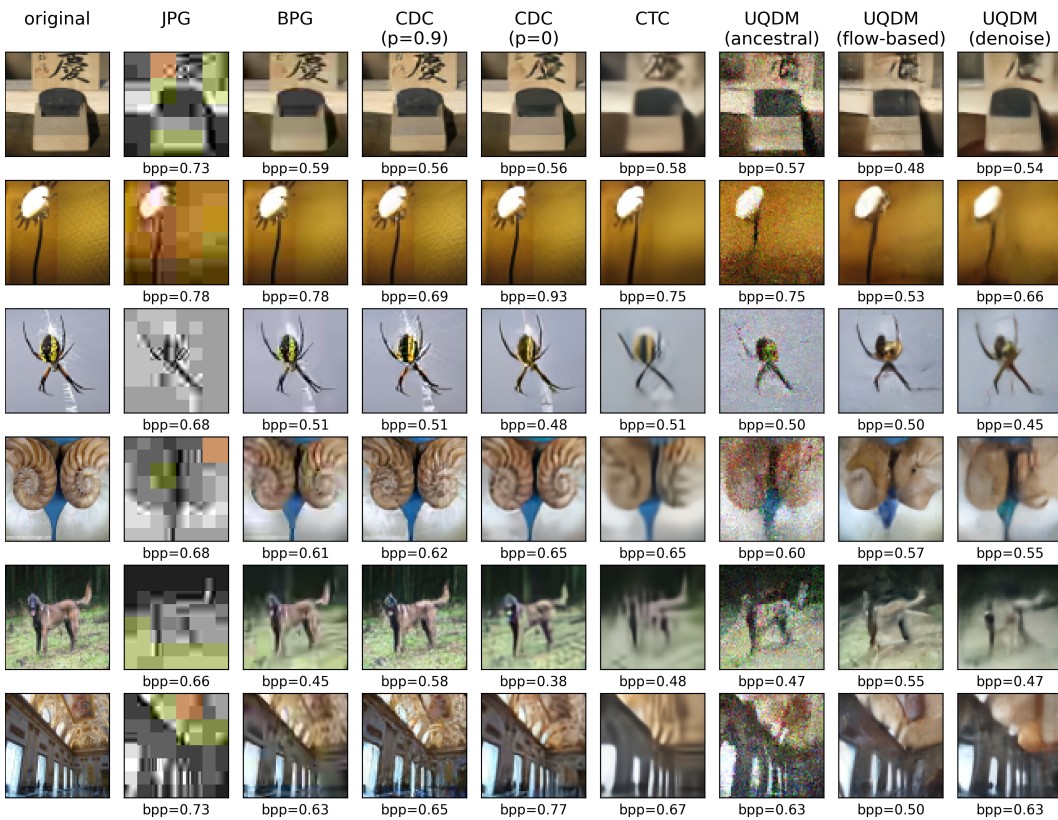

Figure 10: Additional example reconstructions , chosen at roughly similar (low) bitrates.

