# OpenReview forum: "Progressive Compression with Universally Quantized Diffusion Models"
_ICLR.cc/2025/Conference — ICLR 2025 Oral_

### Official Review · Reviewer_evNJ · 2024-10-21

**Soundness:** 3
**Presentation:** 4
**Contribution:** 4
**Rating:** 8
**Confidence:** 4

**Summary:**

### Background:

Ho et al [1] proposed an algorithm that compresses a data sample (e.g an image) down to as many bits as the DDPM’s log-likelihood estimate of that sample. This is very interesting since diffusion models are probably the most advanced image priors ever developed. This compression algorithm has the additional advantage of being naturally progressive, that is, the data is communicated as a sequence of progressively less noisy images. However, a key component of this algorithm, called *relative entropy coding*, has an exponential runtime for gaussian diffusion models. This exponential runtime is a key bottleneck to practical implementation.

### Summary:

This paper introduces a clever solution to a key bottleneck of an otherwise very promising compression algorithm. The authors introduce a new diffusion process which is constructed for computationally efficient relative entropy coding. This solves the exponential runtime problem associated with Gaussian diffusion compression, and allows the authors to implement an actual compression codec, where prior works (Ho et al [1], Theis et al [3]), only calculated the hypothetical performance of this compression algorithm, since it was not practical to implement due to the exponential runtime issue. The authors create new likelihood models based on Kingma’s VDM [2]. The authors evaluate their compression model on three datasets: the toy “swirl” dataset, CIFAR10, and ImageNet64.

**Strengths:**

Paper is well-written and well-motivated. Topic is theoretically interesting and IMO potentially high-impact. The proposed solution is clever, well-explained, and definitely meets the threshold for novelty.

**Weaknesses:**

I have only one major concern: While the authors’ modifications to Kingma’s Variational Diffusion Model (VDM) make relative entropy coding computationally tractable, these modifications also appear to significantly degrade VDM's performance.

For example, the VDM paper included a figure (Figure 3) with non-cherry-picked unconditional samples from their model. IIUC, in Figure 5 of the paper under review, the two bpp=0.00 images generated via ancestral sampling are comparable to the images Kingma et al.’s Figure 3. If this comparison is fair, UQDM has suffered a major degradation in sampling quality compared to VDM.

I am pretty hung up on UQDM’s very low optimal step count. The authors find that their model achieves the best NELBO performance with just 4 or 5 denoising steps. This is counter to the VDM paper which concluded that more steps leads to a lower loss (see Kingma appendix F).

This seems problematic because a major theoretical underpinning of DDPMs is that the ideal $p(x_{t-1} | x_t)$ is well-approximated by a Gaussian as the step size becomes small. But with such large steps, the ideal $p(x_{t-1} | x_t)$ must surely have some very complicated structure which is poorly captured by the author’s parameterization. This might also explain why the authors see such an advantage from learning the reverse-process variance; you just need a lot more parameters to model this distribution accurately. With say T=1,000, I would guess that the fixed-variance version of UQDM would do a better job of modeling the reverse process. Except for some reason, UQDM’s performance degrades with increasing step size.

My gut reaction is that this ultra-small optimal step count seems pathological. But I am open to being convinced that I’m mistaken. What do you think is going on here?

**Questions:**

I see your lossless compression results for the swirl data, but not for CIFAR10 or ImageNet. Would like to see these but they are not critically important.

Theis et al [3] found probability flow based denoising to induce less distortion than ancestral sampling while still preserving realism. Any particular reason you chose ancestral sampling over this flow-based denoising?

If UQDMs and traditional gaussian diffusion models are equivalent in the continuous-time limit, do you think there’s a point at which you could trivially convert a DDPM into a UQDM without retraining it, just by replacing the gaussian noise at each timestep with uniform noise? Or am I misunderstanding something?

## References:

1. Ho: https://arxiv.org/abs/2006.11239
2. Kingma: https://arxiv.org/abs/2107.00630
3. Theis: https://arxiv.org/abs/2206.08889

---

### Official Review · Reviewer_ouDv · 2024-11-04

**Soundness:** 2
**Presentation:** 2
**Contribution:** 3
**Rating:** 8
**Confidence:** 3

**Summary:**

A single diffusion model can be used for data compression at arbitrary bit rates, flexibly trading off rate and distortion, using relative entropy coding (REC). The main drawback is that REC has in general exponential runtime. In this paper, the authors replace Gaussian distributions in the forward process (viewing diffusion models as hierarchical VAEs with a finite number of stochastic layers) with uniform distributions. This allows them to apply universal quantization which leads to cheaper simulation, thereby avoiding the intractability of previous work that was based on Gaussian noise. In addition, the authors show that their diffusion model with uniform noise converges to standard VDM in the continuous-time limit. In practice, this method obtains competitive R-D and R-realism performance, with a single model.

**Strengths:**

- This paper presents an interesting idea with relevant contributions for the community. The focus on improving compression with general-purpose diffusion models is exciting and relevant.
- The theory developed around uniform-noise diffusion, along with its connection to standard Gaussian diffusion in the continuous-time limit, is well-articulated and adds depth to the work.
- The results seem quite promising.

**Weaknesses:**

- **Clarity Issues in Background**: The paper could be clearer, especially in the background section. It would help if it were more accessible to readers who aren't already familiar with probabilistic generative models for compression. For example:
  - It would be useful to give a more intuitive explanation of transmitting a sample from $q$ with close to KL nats, as this is central to the argument about the NELBO corresponding to the lossless coding cost.
  - Some wording is a bit vague, like "using roughly $L_{\mathbf{x}|\mathbf{z}_0}$ nats." This phrasing might make sense to researchers in this specific area, but it could be clearer for others (in particular "roughly" in what sense?).
  - It’s also unclear what exactly the relationship is between REC and universal quantization. My impression is: (1) REC is usually inefficient (with exponential runtime, but exponential in what?), (2) in some specific cases, it may be more efficient, (3) past work has tackled a few of these cases, (4) this paper presents another one of these (which had never been explored before) using a uniform noise channel. I don't know if this is accurate, but in any case the overall picture and context could be a bit more straightforward than it is in the current submission.

- **Structure and Flow**: Wouldn’t it make sense to first define the new diffusion model class with uniform distributions, then explain that this enables universal quantization and more efficient REC? This could better separate the two main ideas: the diffusion model itself ("substituting Gaussian noise channels for uniform noise channels" -- line 185) and its application to compression. If that’s not a sensible structure and my interpretation is completely off, then there may be additional clarity issues that need attention. (I don’t doubt that experts in this exact topic would follow along, though.)

- **Exponential Runtime Explanation**: There are a few statements about REC's exponential runtime and how this new approach is more efficient. But it’s not clear what "exponential" means here or what it’s exponential with respect to. It would help if the paper included: (1) a clear explanation of what exactly is exponential, (2) the complexity of the proposed algorithm (this doesn’t seem to be covered in the main text or appendix), and (3) solid empirical and quantitative support for this efficiency claim (again, this doesn't seem to be mentioned or investigated?).

- **Algorithm Box**: Including a box or diagram for the main algorithm could help make things clearer. It would give readers a more accessible overview and something concrete to reference.

**Minor**:
- Typo on line 214 (missing closing parenthesis in the sigmoid).
- Figure layout could be improved: there’s a lot of whitespace, some figures appear too early relative to where they’re discussed, and the order seems off (e.g., text references Figures 3, 4, 1, 5 in this order, and Figure 2 only appears in the Appendix).

**Overall**: This paper introduces an interesting approach to using diffusion models for flexible data compression, but in my opinion it would significantly benefit from some improvement in clarity and presentation. Making the key concepts easier to understand, reorganizing the content for better flow, and providing detailed support for the claims -- especially regarding complexity -- would be really helpful. Addressing these points will significantly strengthen the paper and enhance its impact.

**Questions:**

- **Appendix C.2 and C.3**: In VDM, the standard architecture typically includes an additional ResNet block in the reverse process (so I would expect 9 blocks here). Could this be a typo?

- **Continuous-Time Equivalence**: Given the equivalence with Gaussian diffusion in continuous time, could this approach be applied to pretrained continuous-time diffusion models by discretizing with uniform distributions? I assume this would be an approximate discretization, as it only matches the mean and variance.

---

> ### Comment · Reviewer_ouDv · 2024-11-30
>
> Thank you for the thorough rebuttal and detailed clarifications. I now have a much clearer understanding of the contributions, and with the improvements in the updated manuscript, I am happy to raise my score to 8.

---

### Official Review · Reviewer_bu5E · 2024-11-06

**Soundness:** 3
**Presentation:** 2
**Contribution:** 3
**Rating:** 8
**Confidence:** 3

**Summary:**

This paper introduces Universally Quantized Diffusion Models, a family of diffusion models utilizing uniform rather than Gaussian noise with the same forward/reverse process tractability (e.g., forward process available in closed form, admit a similar ELBO, etc.) This is achieved by constructing a probabilistic model following the DDIM factorization. The use of uniform noise is motivated in the context of compression, namely to make Relative Entropy Coding (REC) tractable through the use of uniform quantization. This allows the use of progressive coding methods used in prior work exploring compression with diffusion models (Ho et al., Theis et al.) for both lossless and lossy compression. Experiments are presented on a toy dataset of swirl curves and on standard+challenging image benchmarks (CIFAR10, ImageNet 64x64), showing encouraging results compared to standard image compression methods (e.g., jpeg) in the rate-distortion frontier, ditto for other non-neural methods (e.g., bpg) though only at certain rate regimes, and readily outperforming other neural methods such as CTC and CDC.

**Strengths:**

- The paper essentially introduces a formulation of diffusion models using uniform (as opposed to Gaussian) noise, effectively turning diffusion models into a method that through end-to-end optimization directly approximates the quantile function of the data distribution, which can be broadly useful beyond the context of compression.
- The mathematical background and literature review are well-presented.
- Even if encoding and decoding uses significantly more FLOPs compared to common non-neural compression schemes, using more compute for less memory can be useful in applications where storage matters more than reads, motivating research on deep-learning-based compression.
- The method is applicable to both lossless and lossy compression.
- Experiments include both toy examples (swirl dataset) and real image compression tasks on difficult benchmarks standard in the literature (CIFAR10, ImageNet 64x64)
- Results on CIFAR10 and ImageNet 64x64 outperform various common lossy image compression schemes like jpeg, jpeg2000, and at certain rates outperform other compression schemes like bpg.
- Results significantly outperform neural-based methods from prior work such as CTC, CDC.
- The importance of learning variances is demonstrated experimentally.
- Further experimental knowledge is introduced, e.g., that learning the noise schedule does not result in significant improvement in log likelihood optimization.

**Weaknesses:**

The biggest weakness seems to be a lack of comparisons against other diffusion-based lossy compression methods from prior work on the image compression tasks. E.g., DDPM is open sourced so this should be trivial to try. I acknowledge that at least two neural codec baselines (CTC, CDC) are included so this is not a fatal weakness. Still, positive results here would further motivate why we would want to follow the proposed method as opposed to prior ones.

Particularly at low bpp rates, bpg which is efficient even in low-memory settings still outperforms diffusion models. This is not a major weakness, as bigger models could result in lower NELBO across the board. The paper could be strengthened by showing the effects of models of different sizes, even if the largest scales are kept reasonably small.

A similar potential improvement to the paper, especially to discuss the practicality of diffusion-based compression, would be to include quantitative comparisons of FLOPS compared to standard image compression schemes.
There are no illustrations of the progressive coding procedure. This would be extremely helpful to include so the method is easier to grasp and follow.

Another potential improvement would be to include comparisons to other methods that are compatible with Gaussian noise, particularly for lossless compression. For example, it should be possible to compare to bits-back coding, however I acknowledge that this cannot be used to compress individual images (as the paper’s method does) but rather entire datasets, with rate dependent on the number of elements being compressed. Highlighting the benefits of the method, e.g., in a table of desiderata, would be additionally helpful.

Showing more image samples, and showing zoom-ins to help the reader see artifacts across the qualitative comparisons would be very helpful. Expanding the range of rates (bpp) on the samples would also be very welcome.

**Questions:**

- Is the fundamental benefit of REC through universal quantization that it enables both lossless and lossy compression? Or what other reasons would we want to do this as opposed to directly following prior work where tractability is sufficient to show results in lossy compression for the same image datasets? This could be explained more succinctly in the paper.
- Are image compression results with rate > 8bpp important to show in any way?
- For lossless compression (Fig 2), are there explanations or at least intuitions for why more denoising steps can result in worse bpd? This seems like an unexpected result.
- This is likely a question from my own lack of experience in this area: suppose that we define the model to first apply the standard Gaussian erf on the input image (re-scaled dependent on the noise level). Does this not convert Gaussian noise into uniform noise (given that through the inverse erf, more specifically through the quantile function of the Gaussian, we can convert uniform noise into Gaussian noise)? Why does communicating with Gaussian noise remain intractable?

---

### Official Review · Reviewer_bjic · 2024-11-08

**Soundness:** 4
**Presentation:** 4
**Contribution:** 3
**Rating:** 8
**Confidence:** 2

**Summary:**

This paper develops a new form of diffusion model based around universal quantization, which uses a uniform distribution in the reverse process instead of a Gaussian. Previous research made the connection between diffusion and progressive compression via relative entropy coding (REC), but those models used a Gaussian distribution which leads to computational complexity that is exponential in the amount of information being coded / transmitted.

This paper notes that "universal quantization" (Zamir & Feder, 1992) efficiently solves the REC problem in the case of a uniform noise channel. This presents an opportunity for efficient progressive compression with diffusion models if the diffusion process (forward and backward) are reformulated to use uniform noise. This paper does that (Section 3) and the resulting model is evaluated empirically in terms of compression performance using toy data and standard image data sets.

**Strengths:**

The primary strength of the paper is recognizing that diffusion models can be reformulated to use uniform noise, that this allows them to use universal quantization (UQ), and that UQ is an efficient solution to the REC problem for a uniform noise channel.

The authors then derive and implement this model, which they call a "universally quantized diffusion model" (UQDM), and evaluate it empirically to show that the theoretical benefits translate to practice as shown by the RD curve comparisons in Figures 3 and 4. They evaluate two approaches to intermediate reconstruction for progressive compression: ancestral sampling and denoising (Fig. 5) and show, qualitatively and quantitatively the pro/cons of each approach.

An additional strength is the clear and concise presentation of the background information (Section 2). While the basics of diffusion models are now probably familiar to most ICLR readers, relative entropy coding, universal quantization, and the connection between the diffusion process and progressive coding probably isn't.

**Weaknesses:**

Although there are many benefits to the proposed approach, it is not strictly better than previous methods at all bit rates and metrics (PSNR and FID are reported). The comparison is complicated somewhat due to a difference in capabilities, e.g., CDC has better FID than UQDM at low bit rates, but UQDM is progressive while CDC is not. Similarly, UQDM is fairly far behind the (theoretical) quality of VDM, but UQDM provides significant real-world benefits in terms of computational requirements.

In terms of computation and runtime, the benefit of UQDM over VDM is clear (and that's one of the primary points of the paper), but I suspect the practical decode time is still far too slow compared to standard codecs (JPEG, BPG, etc.) to be usefully deployed. This is an issue for much of the neural compression literature, though, so I don't think it should be a big factor in terms of the ICLR acceptance decision (e.g., CDC uses a diffusion-based decoder so is likely also slow in practice compared to JPEG and BPG).

**Questions:**

I'm surprised that BPG does so well in terms of FID around 2.5 bpp (Fig. 4, right), where it even outperforms VDM (T=1000). Is there something to learn from this observation, or does it say something about the limits of VDM?

---

### Meta-Review · Area_Chair_BLc7 · 2024-12-21

**Metareview:**

This paper draws a connection between diffusion models and compression, and proposes method that achieves promising results on image compression. The reviewers unanimously agree that this paper presents significant contributions to the field. I believe that this work points to an important direction and recommend this paper being presented as a spotlight paper.

**Additional Comments On Reviewer Discussion:**

Some concerns raised by reviewers include more thorough experimental comparisons, absolute results and clarity of presentation. The authors did a great job in the rebuttal addressing these, which is reflected in the increased ratings after rebuttal.

---

### Decision · Program_Chairs · 2025-01-22

Accept (Oral)